# An attribution of the low single-scattering albedo of biomass-burning aerosol over the southeast Atlantic

Amie Dobracki[1], Paquita Zuidema[1], Steve Howell[2], Pablo Saide[3], Steffen Freitag[2], Allison C. Aiken[4], Sharon P. Burton[5], Arthur J. Sedlacek III[6], Jens Redemann[7], and Robert Wood[8]

[1]Department of Atmospheric Sciences, Rosenstiel School, University of Miami, Miami, Florida, USA
[2]University of Hawaiʻi at Mānoa, Honolulu, Hawaii, USA
[3]University of California Los Angeles, Los Angeles, California, USA
[4]Earth and Environmental Sciences Division, Los Alamos National Laboratory, Los Alamos, New Mexico, USA
[5]NASA Langley Research Center, Hampton, VA, USA
[6]Brookhaven National Laboratory, Upton, New York, USA
[7]University of Oklahoma, Norman, Oklahoma, USA
[8]University of Washington, Seattle, WA, USA

**Correspondence:** Paquita Zuidema (pzuidema@miami.edu) and Amie Dobracki (amie.dobracki@rsmas.miami.edu)

**Abstract.** Aerosol over the remote southeast Atlantic is some of the most sunlight-absorbing aerosol on the planet: the *in-situ* free-tropospheric single-scattering albedo at the 530 nm wavelength ($SSA_{530nm}$) ranges from 0.83 to 0.89 within ORACLES (ObseRvations of Aerosols above CLouds and their intEractionS) aircraft flights from late August-September. Here we seek to explain the low SSA. The SSA depends strongly on the black carbon (BC) number fraction, which ranges from 0.15 to 0.4. Low organic aerosol (OA) to BC mass ratios of 8-14 and modified combustion efficiency values > 0.975 point indirectly to the dry, flame-efficient combustion of primarily grass fuels, with back trajectories ending in the miombo woodlands of Angola. The youngest aerosol, aged 4-5 days since emission, occupied the top half of a 5 km thick plume sampled directly west of Angola with a vertically-consistent $BC:\Delta CO$ (carbon monoxide) ratio, indicating a homogenization of the source emissions. The younger aerosol, transported more quickly off of the continent by stronger winds, overlaid older, slower-moving aerosol with a larger mean particle size and fraction of BC-containing particles. This is consistent with ongoing gas condensation and the coagulation of smaller non-BC particles upon the BC-containing particles. The particle volumes and OA:BC mass ratios of the older aerosol were smaller, attributed primarily to evaporation following fragmentation, instead of dilution or thermodynamics. The CLARIFY (CLoud-Aerosol-Radiation Interaction and Forcing: Year-2017) aircraft campaign sampled aerosols that had traveled further to reach the more remote Ascension Island. CLARIFY reported higher BC number fractions, lower OA:BC mass ratios, lower SSA yet larger mass absorption coefficients compared to this study's. Values from one ORACLES-2017 flight, held midway to Ascension Island, are intermediate, confirming the long-range changes. Overall the data are most consistent with continuing oxidation through fragmentation releasing aerosols that subsequently enter the gas phase, reducing the OA mass, rather than evaporation through dilution or thermodynamics. The data support the following best-fit: $SSA_{530nm}=0.801+0055*(OA:BC)$ ($r = 0.84$). The fires of southern Africa emit approximately one-third of the world's carbon; the emitted aerosols are distinct from other regional smoke emissions and their composition needs to be represented appropriately to realistically depict regional aerosol radiative effects.

# 1 Introduction

Biomass burning, the largest source of carbon to the atmosphere globally, is fundamental to the Earth's global carbon cycle (Bowman et al., 2009; Bond et al., 2013). The emissions include carbon monoxide, carbon dioxide, methane and carbonaceous aerosols, significantly altering the atmospheric composition over large regions of the globe (Andreae, 2019). This in turn influences all of the gaseous, aerosol and aerosol-cloud interaction radiative forcing terms considered within the IPCC Assessments. Despite the importance of biomass burning events on climate, smoke properties after long-range transport are still poorly characterized. These include the effluents from northern European and Russian forest fires reaching the Arctic basin (Cubison et al., 2011), wildfire smoke from western continental north America observed over Europe (Zheng et al., 2020; Baars et al., 2021), and aerosols from fires in southern Africa reaching south America (Holanda et al., 2020). Without wet or dry scavenging, the aerosol's areal coverage is increased through transport, increasing the aerosol's radiative impact.

Southern Africa region produces approximately one-third of the world's fire-emitted carbon (van der Werf et al., 2010). The global maximum of absorbing aerosol above cloud occurs above the southeast Atlantic (Waquet et al., 2013), a combination that produces a direct radiative warming of the regional climate (Keil and Haywood, 2003; Graaf et al., 2014; Zuidema et al., 2016; Mallet et al., 2021; Doherty et al., 2022). Biomass-burning aerosol (BBA) from this region is unusual for being highly absorbing of sunlight, with SSA values of 0.85 or less at the green wavelength (Zuidema et al., 2018; Chylek et al., 2019; Pistone et al., 2019; Holanda et al., 2020; Taylor et al., 2020; Denjean et al., 2020b; Mallet et al., 2020; Shinozuka et al., 2020; Carter et al., 2021; Brown et al., 2021). More absorbing aerosol will reduce the need for latent heat release as a balance to longwave radiative cooling within the world's energy distribution (Pendergrass and Hartmann, 2012) and alters regional circulation and precipitation patterns (Mallet et al., 2020; Solmon et al., 2021; Chaboureau et al., 2022). While climate models discern an ensemble-mean direct radiative warming, individual models disagree strongly on magnitude and even sign (Myhre et al., 2013; Zuidema et al., 2016; Haywood et al., 2021; Mallet et al., 2021). In addition, the direct aerosol radiative effect estimated from satellites typically exceeds model estimates (de Graaf et al., 2020). That the measured SSAs are lower than what is currently implemented in many models (Shinozuka et al., 2020; Mallet et al., 2021; Doherty et al., 2022), suggests one cause for an underestimated modeled direct radiative warming is a model SSA that is too high.

This study's goal is to examine the optical properties and composition of *in-situ* smoke sampled in the free tropopshere during six flights of the NASA Earth Venture Suborbital-2 ORACLES (ObseRvations of Aerosols above CLouds and their intEractionS; Redemann et al., 2021) deployment, primarily from September, 2016 (Fig. 1). Complementary observations were taken in this region by the UK CLARIFY (CLoud-Aerosol-Radiation Interaction and Forcing: Year-2017) aircraft campaign (Haywood et al., 2021) on Ascension Island (8°S, 14.5°W), from August 17 - September 7, 2017, and the DACCIWA (Dynamics-Aerosol-Chemistry-Clouds Interactions in West Africa) airborne campaign over southern west Africa during June-July 2016 (Knippertz et al., 2015). Both campaigns have already revealed that southern African biomass-burning aerosol (BBA) is highly absorbing of sunlight because the fractional black carbon content is high in both number and mass of total particles (Taylor et al., 2020; Denjean et al., 2020b), with loss of particle coating also contributing (Sedlacek et al., 2022). We further strengthen the attribution to aerosol composition, fire source characteristics and indicators of chemical aging and seek to

place the ORACLES data within the context of these other measurements. The aerosol sampled during the Southern African Regional Science Initiative (SAFARI) campaign (Haywood et al., 2003) held in 2002 near Namibia was less than 3 days of age, while the CLARIFY-2017 campaign over Ascension sampled aerosol approximately a week old (Wu et al., 2020; Taylor et al., 2020). The ORACLES-2016 model-derived age estimates place the flights uniquely within the time record available on chemical aging over the southeast Atlantic.

The paper is organized as follows: Section 2 describes the Methodology, including description of the flights and relevant details about the datasets, with the more technical details relegated to the Supplement. Section 3 presents the chemical characterization, including the aerosol age estimates. Section 4 discusses the chemical optical and physical properties of the smoke plumes, including the likelihood of brown carbon. Section 5 investigates how the organic aerosol component varies between the flights and if OA differences can be explained as a aging process, with Section 6 incorporating a comparison to measurements made at Ascension. Section 7 provides a summary and discussion.

## 2   Methodology

Six flights were selected, based on the availability of at least 20 minutes of organic aerosol (OA) masses exceeding $>20$ $\mu$g m$^{-3}$ (a threshold justified in section 2.4.2), at altitudes above 1.5 km and relative humidities (RH) $< 80\%$. The latter is applied to reduce the likelihood of aqueous-phase reactions. The flights occur within 30 days of each other in the seasonal cycle, spanning August 31 through all of September, to preferentially select for similar composition of the fire source emissions. Five flights come from 2016 and one from 2017, with their tracks shown relative to the satellite-derived above-cloud aerosol optical depths for September 2016 in Fig. 1. Spatial maps of the aerosol forecasts used to guide the flight planning are shown overlaid with each flight's track, along with measured OA mass concentrations and a model-estimated aerosol age based on CO tracers (explained further in Section 2.2) on individual altitude-latitude flight track projections in the Supplementary Figs. S1-S2.

More description of the flights is followed by descriptions of the datasets that are more central to the results (listed in Table 1), with the more technical details provided in the Supplement.

### 2.1   Flight Description

The aircraft flew along a routine southeast to northwest track on three flights (31 August, 4 and 25 September of 2016), and performed three target-of-opportunity flights sampling more aerosol-rich locations (6 and 24 September of 2016, 31 August 2017), all shown in Figs. S1-S2. The flight tracks make clear that the aircraft sampled widely, but never near the fire emission sources, with the 9/24/2016 flight coming closest (Fig. 2). The aerosol spatial distribution is strongly influenced by the strength of the free-tropospheric easterly winds, with the aerosol either constrained to be near the coast when the winds are weak, or elongated zonally along 10°S when the winds are strong. On the 24-25 September 2016 and 31 August 2017 flights, the zonal easterly winds exceeded 6 m s$^{-1}$ along $\sim 10°$S at altitudes between 3-5 km, forming a wind isotach known as the African Easterly Jet-South (Nicholson and Grist, 2003; Adebiyi and Zuidema, 2016). Overall, September 2016 was climatologically representative (Ryoo et al., 2021), with more synoptic detail on individual flight days available in Ryoo et al. (2022).

An aerosol forecast model was used to seek out smoke layers to sample during the target-of-opportunity flights. The six flights intersect aerosol of different ages, but all model-estimated ages exceed 4 days. Table S1 lists all of the ORACLES-2016 flights and includes comments on their flight pattern, the number of seconds with OA>20 $\mu$g m$^{-3}$, and other selection considerations. We highlight two flights further: one from 9/24/2016, because it sampled a thick, younger smoke plume as close as possible to a fire emission source (Fig. 2), and the 9/31/2017 flight, which sampled an aerosol layer of significant mass and extremely stable OA:BC halfway between Ascension and the African continent, and helps connect interpretations of ORACLES to CLARIFY aerosol characteristics. The focus on these two flights has been added since (Dobracki et al., 2022). Optical properties were primarily examined for data from level legs, for which further time averaging could reduce the measurement uncertainty. Table S2 provides flight dates, location, time span and altitude of the level leg data we analyze further, ranging four to ten minutes in length.

## 2.2 Determination of physical aerosol age

Model-released tracers tagged to CO at the fire source for each day of the campaign's operational two-week aerosol forecast, made using the Weather Research Forecasting - Aerosol Aware Microphysics (WRF-AAM) Model (Thompson and Eidhammer, 2014), were used to estimate the physical age of the aerosol. The regional model has a12-km spatial resolution and encompasses a domain (41°S-14°N, 34°W-51°E) sufficiently large to capture almost all contributing fires (Saide et al., 2016). The model is driven by the National Center for Environmental Prediction Global Forecasting System (NCEP GFS) meteorology, using daily smoke emissions from the Quick Fire Emissions Dataset (Darmenov and da Silva, 2013) released into the model surface layer. These are advected thereafter according to the model physics, with their spatial distribution constrained near real-time with satellite-derived optical depths. Most fires in southern Africa occur during the day, and the satellite constraint captures this diurnal cycle. The model fire emissions rely on a burned-area product of 500 m spatial resolution from the Moderate Resolution Imaging Spectrometer (Giglio et al., 2006). This may miss up to $\sim 40\%$ of the total burned area coming from smaller fires (Ramo et al., 2021). Larger fires, with more protected cores, contribute more to the emissions reaching higher altitudes for the boreal northern hemisphere (Martin et al., 2010). We are unsure how well this same vertical selection applies for the smaller agricultural fires of southern Africa, although stronger zonal winds aloft will aid lofting of the smoke underneath (Adebiyi and Zuidema, 2016). The tracer-derived estimates tend to keep the smoke emissions near the surface until the aerosol is eventually carried aloft, shown for 24 September 2016 in Fig. S3. The time lag allows the emissions from nearby fire sources to mix, homogenizing local differences in, e.g., grass versus leaf-litter, moisture content, and surface burn history. Conditions can nevertheless still change from day to day.

Backtrajectories based on the HYSPLIT model (Stein et al., 2015), also driven by the NCEP GFS meteorology, further illuminate the pathway the BBA traveled prior to its sampling on 24 September 2016 (shown later in Fig. 12d). These backtrajectories reach the location of the fire emission source after approximately three days, at higher altitudes than where the aerosol was sampled. The aerosol age estimate is younger than the WRF-AAM aerosol forecasts would suggest. This is because the time needed for the aerosol to travel from the surface to the higher altitude is not accounted for in the HYSPLIT age

estimate. The vertical advection timescale is highly model-dependent on boundary layer parameterizations and we merely note the difference in the two age estimates here.

## 2.3 Modified combustion efficiency

CO and carbon dioxide ($CO_2$) are used to infer fire emission conditions through the modified combustion efficiency (MCE)
metric (Collier et al., 2016; Yokelson et al., 1997):

$$MCE = \frac{\Delta CO_2}{\Delta CO + \Delta CO_2} = \frac{1}{1 + \Delta CO / \Delta CO_2} \qquad (1)$$

An MCE of 0.9 marks the 50% threshold between flaming and smoldering combustion (Akagi et al., 2011), a threshold that is largely insensitive to fuel type (May et al., 2014). Higher values of MCE (>0.9), more associated with flaming combustion, preferentially produce more BC, whereas an MCE < 0.9 is more typical of a smoldering fire that emits more organic aerosol for the same amount of fuel (Yokelson et al., 2009; Vakkari et al., 2018). A regression is used to estimate the $\Delta CO / \Delta CO_2$ with $\Delta CO$ and $\Delta CO_2$ calculated from the measured CO and $CO_2$ amounts, in moles, relative to background values. Adopted background values were 65 (77) ppbv for CO, and 397 (404) ppmv for $CO_2$, in September 2016 (August 2017), based on measurements in the free troposphere taken above the smoke plumes ($\sim$7000 m).

## 2.4 Aerosol Composition

### 2.4.1 Black carbon

The rBC size distributions are derived from a 4-channel single particle soot photometer (SP2, Droplet Measurement Technology) deployed by the Hawaii Group for Environmental Aerosol Research HiGEAR in 2016, and an 8-channel SP2, of which only the incandescent channels were functional for the August 31, 2017 flight, deployed by Art Sedlacek of Brookhaven National Laboratory. No scattering data are available, precluding information on coating thicknesses. The intensity of laser-induced incandescent emission at 1064 nm can be quantitatively related to the mass of the refractory black carbon (rBC) particles for mass-equivalent diameters between approximately 0.08-0.5 $\mu$m. The SP2 was calibrated using fullerene soot using effective density estimates from Gysel et al. (2011). Calibration uncertainty dominates the nominal mass uncertainty of $\pm$ 17% (Laborde et al., 2012).

Size distributions by number and mass (assuming a BC density of 1.8 g cm$^{-3}$ (Bond and Bergstrom, 2006)) for the level legs detailed in Table S2 indicate particle diameters remain well below the upper SP2 detection limit (Fig. 3). Lognormal fits help visualize a drop-off in detection efficiency for diameters < 0.08 $\mu$m for the samples weighted towards the smaller sizes (Fig. S4, for 9/24/2016 and 9/25/2016). The SP2 size range nevertheless captures almost all of the black carbon mass, close to the 99% value reported for CLARIFY (Taylor et al., 2020). rBC total number concentrations vary between 530 to 1370 cm$^{-3}$ (Fig. 3a), and undercounting of the mass and number through coincidence is estimated to be less than 3% based on Taylor et al. (2020). Throughout, we use BC to refer to the SP2-derived refractory black carbon, following other literature, although the two are not entirely the same (Petzold et al., 2013).

Ratios of $\frac{\Delta BC}{\Delta CO}$, reducing to $\frac{BC}{\Delta CO}$ because the background BC is zero in clean conditions, serve to assess homogeneity of the aerosol composition at the emission source. The ratios are non-dimensionalized by using the ideal gas law at standard temperature (273K) and pressure (1000 hPa) to convert the CO concentrations from ppb to ng m$^{-3}$.

## 2.4.2 Aerosol Mass Spectrometer measurements

HiGEAR operated an Aerodyne High-Resolution Time-of-Flight Aerosol Mass Spectrometer (HR-ToF-AMS, referred to as AMS), building on previous experience in the southeast Pacific (Yang et al., 2011; Shank et al., 2012) and the Arctic (Howell et al., 2014). This measured masses of organic aerosol (OA), nitrate (NO$_3$), sulfate (SO$_4$) and ammonium (NH$_3$). Chloride, a small component of the total aerosol mass in the free troposphere, was not considered because of its inconsistent ionization signature. The native time resolution is approximately five seconds, with the data interpolated onto a one-second temporal grid to facilitate integration with other datasets. The overall uncertainty in the reported aerosol mass concentrations is estimated at 33% to 37% at a one-minute time resolution, based on Bahreini et al. (2009), generating a combined uncertainty in the OA to BC mass ratio of close to 40% (since the background OA in clean conditions is also zero, $\frac{\Delta OA}{\Delta BC}$ reduces to OA:BC).

Means over the level legs listed in Table S2, further reduce OA mass uncertainties to 19%-10 %, and to 25%-19% for the OA:BC mass ratio. The instrument inter-comparison flight with CLARIFY sampled a clean troposphere but a polluted boundary layer (BC of $\sim 300$ ng m$^{-3}$), during which the ORACLES OA and nitrate mass concentrations were 80% of those measured by the UK plane, within each other's standard deviations (Barrett et al., 2022).

A threshold of 20 $\mu$g m$^{-3}$ is applied to the OA mass to select for the heart of the smoke plumes. This is one approach to minimizing dilution effects, by which OA evaporates through mixing with cleaner environmental air (e.g., Hodshire et al., 2021). This threshold was selected based on when a stabilization of the OA:BC mass ratio occurs as a function of the OA mass concentrations (Fig. 4). The OA:BC mass ratio is significantly less for air with OA>3 $\mu$g m$^{-3}$ than for air with OA>20 $\mu$g m$^{-3}$, particularly for younger aerosol (Fig. S5), consistent with evaporation through dilution. The choice of threshold is inherently arbitrary, and some analysis is repeated using an OA>10 $\mu$g m$^{-3}$ threshold to make sure our findings are not sensitive to the choice of OA mass threshold. The BC:$\Delta$CO and OA:BC ratios are shown for individual flights as a function of *f44* for both thresholds in Figs. S11-S14 in the interest of full documentation. We also account for dilution effects by normalizing OA with respect to BC or $\Delta CO$, two quantities that do not change with dilution. We stress that the aerosol plumes over the southeast Atlantic, termed 'rivers of smoke' within Swap et al. (2003), are typically much larger and homogeneous than the fire plumes sampled in the western northern hemisphere, which are often linked to named, individual fires and sampled close to the source. The plume sampled on 8/31/2017, halfway to Ascension Island, is an example of a wide, homogeneous smoke plume (Fig. 3). OA mass concentrations often remained highly stable over level legs (see Fig. S2, bottom row, for an example). Further justification for a threshold is that the OA mass uncertainty is smaller at higher signal-to-noise ratios. Additionally, model-observational disparities in the smoke plume locations have less impact on further aging-related analyses if based on the plume centers. For the same reason we exclude aerosol with physical ages > 10 days as the model skill in predicting smoke age is likely poorer by then.

Other AMS measurements include the fraction of the OA mass spectrum signal at *m/z 44, 43* and *60* relative to the total OA mass concentration, termed *f44*, *f43*, *f60*, and hydrogen (H), oxygen (O), and organic carbon (OC). *f44*, indicates the presence of the $CO_2^+$ ion, a form of oxidation resulting from chemical aging (Canagaratna et al., 2015). *f43* indicates the presence of $C_2H_3O+$ and $C_3H_7^+$, also representative of oxygenated OA. *f60* indicates $C_2H_4O_2$, a fragment of levoglucosan and a known tracer for biomass burning aerosol (Cubison et al., 2011). Elemental analysis, yielding H, O and OC rely on algorithms from

Aiken et al. (2007). The calibration constants differ between the two years but this change does not quantitatively impact any differences shown here.

## 2.5   Determination of organic/inorganic nitrate contribution

Farmer et al. (2011) provide an approach for estimating the contribution to the total nitrate signal from organic nitrate (ON) using the $NO^+:NO_2^+$ ratio, building on the observation that organic nitrates typically fragment into larger proportions of $NO^+$

than do inorganic nitrates. In their study, organic $NO^+$ ratios vary between 1.8 to 4.6 for different organonitrates, compared to 1.5 for $NH_4NO_3$. Their Equation 1, reproduced below, provides an estimate of the ON fraction that can be readily applied to the ORACLES AMS data, assuming enough ON is present that it can be resolved. The success of this approach also assumes that the inorganic nitrates capable of providing a large $NO^+$ ratio, such as mineral nitrates, are not present. Both assumptions are justified for the southeast Atlantic (SEA) free troposphere.

$$X(ON\%) = \frac{(R_{obs} - R_{NH_4NO_3})(1 + R_{ON})}{(R_{ON} - R_{NH_4NO_3})(1 + R_{obs})} \qquad (2)$$

$R_{obs}$ is the ORACLES *m/z* ratio of ion fragments 30 to 46, $R_{NH_4NO_3}$ is the ionization efficiency (IE) calibration-derived ratio (1.26 for 2016 and 1.545 for 2017) and an $R_{ON}$ value of 3.41 is a reference ratio based on the average fragmentation pattern into the $NO^+:NO_2^+$ ratios for the OIA-HN, OIA-CN and OIA-olig standards evaluated within Table S1 of Farmer et al. (2011). The inorganic nitrate (IN) fraction is 1-ON. We use this approach to estimate the inorganic nitrate (IN, primarily $NH_4NO_3$)

fraction, keeping in mind that it is an indirect inference. The CLARIFY campaign relied on assessing *m/z* 30 and *m/z* 46 (primarily $NO^+$ and $NO_2^+$) to assess the organic to inorganic nitrate contribution.

## 2.6   Aerosol sizing

Total aerosol number concentrations from a Condensation Particle Counter (CPC; TSI 3010, marked 'ACN' in Fig. S8) establish the fraction of BC-containing particles. The CPC counter applies a size threshold of 10 nm with no upper bound. Aerosol

size distribution measurements rely on a long differential mobility analyzer (LDMA; heavily modified from a TSI 3071A). The aerosol sizing from an LDMA and a thermal DMA (TDMA) are preferred to those from a Ultra-High Sensitivity Aerosol Spectrometer (UHSAS; DMT) because of UHSAS sizing uncertainties (Howell et al., 2021). A correction for a known undersizing by the UHSAS, put forward in Howell et al. (2021), is evaluated in Fig. S7, in which the LDMA and UHSAS median diameters are compared for the level-leg plumes (Table S2) with OA>20 >20 $\mu$g m$^{-3}$. Two particle populations emerge: one with diam-

eters between 0.15-0.2 $\mu$m that is more likely to contain black carbon (see Fig. 3), and another with median diameters < 0.1 $\mu$m, speculated to consist mostly of OA. For the larger particles containing BC, the UHSAS correction reduces the undersizing

bias to 15% compared to LDMA median diameters > 150 nm, but for the smaller particles that are less likely to contain BC, the UHSAS particle sizes are now overcorrected. For this reason, and because a particle cavity aerosol spectrometer probe (PCASP) underperformed, we only show aerosol sizes based on the LDMA data. Analysis duplicated using UHSAS data did not contradict our findings.

The LDMA measures mostly singly-charged particles between 10-550 nm in mobility diameter, with multiply-charged particles occurring at diameters > 200 nm (Howell et al., 2021). The inversions include a size-dependent charging efficiency that accounts for the multiple charges and for size-dependent losses (Zhou, 2001). The HiGEAR LDMA operated in a scanning mode at ambient temperature and pressure, drawing in desiccated air (RH<30%) from an aluminum lagged-aerosol grab chamber for 60 seconds. The total number uncertainty is estimated to be ±30 % due to errors in sizing of non-spherical particles along with uncertainties of flow rate. All size distributions and concentrations are corrected to STP (T=25˚C, p=1000 hPa).

The black carbon core mass-median diameter is used to infer fire conditions at the source. Larger BC sizes correspond to more woody fuels than grasses in Holder et al. (2016), while larger BC cores are associated with more flaming conditions in Pan et al. (2017), attributed to less oxygen reaching the interior flame zone. An estimate of the fraction of total particles containing black carbon (FrBC) is also constructed from the total number of SP2-derived BC particles divided by the total CPC particles.

## 2.7 Optical Measurements

Scattering from all particles is measured continuously by a nephelometer (TSI 3563) at the (450, 550, 700) nm wavelengths ($\lambda$), from which scattering coefficients ($\sigma_s$) are retrieved. The spectral light absorption coefficients ($\sigma_a$) of the total aerosol are estimated from Particle Soot Absorption Photometer (PSAP; Radiance Research) measurements at the (470, 530, and 660) nm wavelengths. The nephelometer scattering measurements are interpolated to the PSAP wavelengths. The extinction (scattering+absorption) and absorption measurements compare well at the blue and green wavelengths to the more sophisticated measurements made by the CLARIFY EXtinction SCattering and Absorption of Light for AirBorne Aerosol Research (EXS-CALABAR) instrument (Davies et al., 2019; Barrett et al., 2022). More detail on the algorithmic treatment of the filter-based measurements is provided in Section 7 of the Supplement.

The absorption Ångström exponents (AAE) are calculated from the linear fit of $\log(\sigma_a)$ to $\log(\lambda)$. The mass absorption cross-section at 660 nm ($\text{MAC}_{BC,660}$) is based on $\sigma_{a,660}$ divided by the BC mass concentration. Following Carter et al. (2021), we also evaluate the MAC relative to the BC+OA mass concentration at $\lambda$=470 nm ($\text{MAC}_{BC+OA,470}$), to assess absorption contributions from both OA-induced brown carbon as well as other wavelength-dependent absorbers (Zhang et al., 2022). The single-scattering albedo is examined at 530 nm ($\text{SSA}_{530}; = \frac{\sigma_{a,530}}{\sigma_{s,530}+\sigma_{a,530}}$) to support comparisons to other published values.

## 3 Chemical composition and age distribution within the six flights

The mean submicron mass fractions of the six flights combined are 66% OA, 10% nitrate ($NO_3$), 11% sulfate ($SO_4$), 5% ammonium ($NH_4$), and 8% BC, with the masses for each species and flight in Fig. 5, thresholded for OA > 20 g m$^{-3}$. Flight-

mean submicron mass totals typically exceed 35 $\mu$g m$^{-3}$. This is much more than measured in the free troposphere above Ascension during CLARIFY (Wu et al., 2020), although the OA mass fraction during CLARIFY still remained $> 50\%$ of the total aerosol mass.

Fig. 6 provides an overview of the *f44*, OA to BC mass ratio, model-derived time since emission (age), MCE, non-dimensionalized $\frac{BC}{\Delta CO}$ ratios and ozone values for each flight. *f44* flight-mean values range from 0.18 to 0.22, on par with *f44* values of Asian/Siberian smoke after a two-week transport to Alaska (Cubison et al., 2011). They are also similar to CLARIFY values (Wu et al., 2020), suggesting a maximum *f44* value of $\sim 0.22$ for this aerosol regime. The *f44* values indicate highly-oxidized aerosol but their range may still contain information on the relative aerosol age: the lowest flight-mean *f44* value from the 9/24/2016 flight, corresponds to the youngest aerosol (Fig. 6c), although the other flight-mean aerosol ages since emission do not correlate well to *f44*.

Flight-mean OA:BC mass ratios range from 7 to 13. MCE values are above 0.97 for each flight. These clearly indicate flame-efficient fires (Collier et al., 2016; Zhou et al., 2017), whose emissions can also more easily reach higher altitudes than can emissions from smoldering fires (Kondo et al., 2011). This may explain why the ORACLES-2016 MCE values exceed the September-mean estimate of $\sim 0.89$ from a source emission-based model (Zheng et al., 2018). Mean non-dimensionalized $\frac{BC}{\Delta CO}$ ratios vary between 0.007 to 0.011, with a minimum on 24 September. These ratios are among the highest surveyed in the literature (Table 2). Overall, $\frac{BC}{\Delta CO}$ ratios do not increase with increasing MCE as expected based on Kondo et al. (2011), but this likely reflects our study's small range of MCE values, for which Vakkari et al. (2018) also do not find a correlation. The mean $\frac{BC}{\Delta CO}$ values hint at a decrease throughout September, with the flight-mean OA:BC mass ratios also increasing to 13 later in September. The flight-mean ozone levels range from 80-105 ppbv, possibly decreasing as September progresses. Flights with more ozone correspond to flights with lower MCEs: less flaming fires will emit more ozone along with more OA. The changes over the course of the month are consistent with more combustible fuel being ignited earlier (Eck et al., 2013), but none of the trends are statistically-insignificant.

We interpret the high MCE values to reflect a large contribution from dry and dead grasses, rather than green grass or more woody materials, for the following reasons. MCE varies inversely with the moisture content for grasses (Korontzi et al., 2003), with leaf litter and woody fuels tending to dry more slowly than do grasses. For this reason woody fuels are more prone to smoldering than flaming combustion. The burning of dry grass produces relatively low emissions of carbon monoxide (Scholes et al., 1996) and higher emissions of black carbon than do agricultural or woodland fires (Andreae, 2019), elevating the $\frac{BC}{\Delta CO}$ ratios, as seen here (Table 2). That the $\frac{BC}{\Delta CO}$ values measured at offshore locations exceeds those measured previously over land (Table 2) could be because emissions from more intense, larger, flaming fires can more easily reach higher altitudes (Martin et al., 2010; Holder et al., 2016), where they can be dispersed further afield through the stronger winds aloft.

Daily maps of fire locations for the flight days (not shown, see Redemann et al. (2021) for the monthly-mean distributions) indicate the BBA sources are primarily fire emissions from miombo woodlands, which contain a significant fraction of savanna grasses and some agricultural fields (Shea et al., 1996; Christian et al., 2003; Korontzi et al., 2003; Vakkari et al., 2018; Huntley, 2019), distributed over a broad geographic region encompassing Angola, Zambia and the Congo. The miombo shrubbery is fire-adapted and less likely to burn than the grass. A survey of the published emission factors for the vegetation types typical

of southern Africa - savannahs, grasslands, agricultural fields, and at times tropical forest indicates that the high $\frac{BC}{\Delta CO}$ ratios reported in Table 1 and Fig. 6 are primarily representative of grass fires. Overall, these metrics indicate aged, oxidized aerosol emanating from flame-efficient fires, without any strong outliers amongst the flights (flight-mean $\frac{BC}{\Delta CO}$ ratios vary from 7-11$*10^{-3}$), typical values for grasslands and savannahs (Janhäll et al., 2010; Vakkari et al., 2018).

Southern African fires can still produce significant near-source secondary organic aerosol (SOA), depending on the burning conditions (Vakkari et al., 2018; Pokhrel et al., 2021). The comparison of *f44* to *f43* for all the flights (Fig. 7a) indicates a mixture of semi-volatile and low-volatile oxygenated organic aerosol (Ng et al., 2011). A PIKA analysis reveals the dominant peak at *f43* is from $C_2H_3O^+$, representative of oxygenated organic aerosol (Ng et al., 2011). *f60* values are relatively constant and below 0.005 (Fig. 7b), and *f44* values lie between 0.2 and 0.22. Chamber studies report lifetimes for *f44* and *f60* of approximately 20 days and 10 hours, respectively (George and Abbatt, 2010; Hodshire et al., 2019), but little change is evident in *f44* after 6 days since emission (Fig. 7c), with *f44* values of 0.2-0.22 also reported at Ascension (Wu et al., 2020), suggesting a steady state has been reached.

The H:C versus O:C mass ratios occur close to the -1 slope line (Fig. 8; based on vanKrevelen (1950)), also inferred at Ascension (Wu et al., 2020). This slope relationship is common to many laboratory and field studies (Heald et al., 2010), with the narrow distribution, particularly within individual flights, suggesting either a limit to the number of oxidation pathways and molecular structures, or, a dominant few. Most of the oxidation states (OS), defined as 2*O:C-H:C (Kroll et al., 2011), lie between -0.2 and 0.5, which Kroll et al. (2011) categorize as "aged" and semi-volatile oxygenated OA (OS between -0.5 to 0). Only the 31 August, 2017 flight has some aerosol that is oxidized enough to be considered low-volatile (OS > 0.5). We are only able to report the end product of the aerosol chemical properties, and different SOA precursors may also contribute to the range of the observed H:C and O:C ratios (Jimenez et al., 2009; Ng et al., 2011). Nevertheless, Kroll et al. (2009) show aerosol with O:C > 0.4 are dominated by fragmentation pathways, in which further oxidation occurs through the loss of a carbon atom (as opposed to functionalization, which adds an oxygen atom). Fragmentation generates relatively small changes in H:C. The fragmentation process releases small amounts of volatile aerosol and we speculate this pathway is suggested by Fig. 8 for the continuing oxidation of ORACLES-2016 BBA.

Flight-mean O:C mass ratios range between 0.61 to 0.69 for the 2016 flights, with small within-flight standard deviations (0.03-0.06, not shown). Overall, the average (± standard deviation) plume values of H:C, O:C, and the organic-aerosol-to-organic-carbon mass ratio (OA:OC) are 1.2 ±0.1, 0.7 ±0.1, and 2.2±0.1, respectively, over all six flights. The OA:OC mass ratio, a measure of the oxygen content that is useful for model evaluation (Hodzic et al., 2020; Lou et al., 2020), are on par with measurements from the Atmospheric Tomography (ATom) campaign made in the same region (Hodzic et al., 2020) and during CLARIFY (Wu et al., 2020). The mean value of 2.2 is substantially higher than common model-applied values of 1.4-1.8 (Aiken et al., 2008; Tsigaridis et al., 2014; Hodzic et al., 2020) and of primary near-source OA:OC ratios of 1.6 (Andreae, 2019).

## 4 Optical and Physical Properties

Here we discuss relationships between the aerosol optical properties to their chemical and physical composition, and examine their spatial distribution, using the more statistically-robust level-leg mean ($\pm$ standard deviations) values (Figs. 9-11).

### 4.1 Mean Relationships

The bulk mass absorption coefficients ($MAC_{BC,660}$) decrease with the estimate of the fraction of particles containing black carbon (FrBC) for the ORACLES-2016 flights (Fig. 9a), with the SSA values decreasing more systematically with FrBC for all four flights (Fig. 9b). The BC-containing particle fraction varies from 0.2 to 0.4, more than the 0.1-0.2 range shown for July south of remote western Africa (Denjean et al., 2020b), and less than the 0.3-0.45 range at Ascension (Taylor et al., 2020). The total particle number was drawn from the full aerosol size distribution within Denjean et al. (2020b), and by a PCASP (0.1-3 $\mu$m) at Ascension. These size ranges are comparable enough to support the comparison across the three campaigns.

Independent electron microscopy on 2017 filter samples found that almost all BC is at least partially coated, meaning the BC particles are dominated by internal mixing (Dang et al., 2022). Nevertheless, the majority of particles cannot include BC, since FrBC<0.5. As the fraction of BC-containing particles increase, the bulk OA:BC mass ratio tends to decrease. $MAC_{BC,660}$ ranges from 9-12 $m^2$ $g^{-1}$, and $MAC_{BC+OA,470}$ from 13-18 $m^2$ $g^{-1}$, corresponding to absorption enhancement factors of 1.2-1.6 (1.7-2.4) at the 660 nm (470 nm) wavelengths, assuming an MAC of 7.5 $m^2$ $g^{-1}$ for uncoated black carbon (Bond and Bergstrom, 2006) (and greatly exceeding the MAC value of 6.25 $m^2$ $g^{-1}$ for strongly light-absorbing carbon (Bond and Bergstrom, 2006)). The mean $MAC_{BC,660}$ of 10.8 $m^2$ $g^{-1}$ is slightly higher than the median value of 9.3 $m^2$ $g^{-1}$ reported in Carter et al. (2021), likely because the BC-enriched 31 August 2017 flight contributes strongly to the mean value reported here. Median LDMA-inferred particle diameters range from 120-210 nm, with no clear relationship to $MAC_{BC+OA,470}$. This indicates the absorption enhancements are governed more by composition than particle size, similar to Denjean et al. (2020a) for June-July BBA close to the near-equatorial African coast. The 8/31/2017 flight, for an FrBC of 0.3, has a higher $MAC_{BC,660nm}$ (by $\sim 2$ $m^2$ $g^{-1}$), lower OA:BC mass ratio, larger LDMA-inferred particle and mass-median BC core size, compared to values from the 9/6/2016 flight of comparable FrBC. The larger BC core size for 8/31/2017 may come from a woodier fuel, supported by backtrajectories emanating from further north (not shown). Woodier material has been shown to generate larger BC sizes irrespective of MCE within Holder et al. (2016).

The single scattering albedos (SSA) at $\lambda$=530 nm range from 0.83 to 0.89, consistent with the ORACLES-2016 mean SSA of 0.86 (inter-quartile range of $\sim 0.028$) based on all the flight data (Pistone et al., 2019). These SSA values are lower than previously documented *in situ* values over land or coastal (Haywood et al., 2003; Formenti et al., 2003; Dubovik et al., 2002), on par with AERONET September-mean values at Mognu (Eck et al., 2013),and higher than those reported at Ascension Island (Zuidema et al., 2018; Wu et al., 2020). An SSA best-fit regression on OA:BC provides a straightforward connection between the BBA chemical and optical properties: $SSA_{530}$=0.801+0.0055*(OA:BC) (Fig. 10a, correlation coefficient $r$ of 0.84). The dependence on BC:TC (total carbon) following Brown et al. (2021) is: $SSA_{530}$=0.929-0.389*(BC:TC) (Fig. 10b; $r$=-0.79, where TC=BC+organic carbon (OC) and OC is estimated from OA:OC=1.26*O:C+1.18 (Aiken et al., 2008).). The

dependence on BC:TC is not as pronounced as in Brown et al. (2021) primarily because our dataset has a smaller SSA range, with no SSAs > 0.9. The variance in SSA is explained slightly better by OA:BC than BC:TC.

## 4.2 Is there evidence of brown carbon?

Taylor et al. (2020) place an upper estimate of 11% on shortwave absorption by brown carbon (BrC) at 405 nm wavelength by the time the BBA plume reaches Ascension Island. Zhang et al. (2022) indicate that other non-BrC materials such as iron oxides absorb sunlight over the southeast Atlantic, so that BrC may contribute even less than < 10% of the total absorption at sub-500 nm wavelengths. By four days since emission, the primary organic aerosol has mostly converted to secondary OA (SOA), which typically absorbs little light (Bond and Bergstrom, 2006; Laskin et al., 2015). Nevertheless, if oxidation can continue to produce new chromophrores (O'Brien and Kroll, 2019) that absorb differently based on wavelength, that could be interpreted as SOA-induced BrC. Ozone Monitoring Instrument UltraViolet Aerosol Index values do suggest OA-produced brown carbon should be present east of the prime meridian (Carter et al., 2021), however. Laboratory studies find more BrC absorption for lower OA:BC mass ratios (Saleh et al., 2014; Holder et al., 2016; McClure et al., 2020), because more intense fires also produce more primary OA and BrC. One important difference is that the reported primary OA fraction and AAEs are much higher within Saleh et al. (2014) than we would expect over the southeast Atlantic.

Motivated by Carter et al. (2021) we examine if distance from the continent has a detectable influence on the absorption Ångström exponents (AAE) calculated over the 470-660 nm wavelength range for the level leg data (Fig. 11a), recognizing that the 470 nm wavelength may already be too long to be responsive to additional absorption by OA-produced BrC (Zhang et al., 2022). The AAEs span 1.1-1.3 south of 8°S irrespective of distance from the coast, and are close to one further north for the more remote 8/31/2017 flight (Fig. 11a). Such AAE values typically indicate a lack of BrC (e.g., Forrister et al., 2015). AAE is weakly positively correlated with OA:BC ($r = 0.27$; not shown), but the relationship is statistically insignificant. Brown carbon absorption is also assessed using $MAC_{BC+OA,470}$ following Carter et al. (2021) (Fig. 11b). These range from 0.94-1.2 m$^2$ g$^{-1}$ south of 8°S (with one exception) to 1.4-1.7 m$^2$ g$^{-1}$ further north. $MAC_{BC+OA,470}$ is anticorrelated with OA:BC ($r$= -0.86). Although consistent with Saleh et al. (2014), the small sample size, dominated by one flight north of 8°S with less OA:BC, precludes much interpretation. We primarily conclude a lack of a longitudinal dependence, although the sample size is too small to say this with confidence. Other work has found that co-emitted sulfate can contribute to increasing overall absorption (Christian et al., 2003), but we do not find a correlation between either MAC estimate and the sulfate fraction (not shown).

## 5 Is there evidence for ongoing loss of organic aerosol? 24 September 2016 case study

The data for the youngest aerosol, aged 4-5 days since emission, stems from the 9/24/2016 flight (Fig. 2). Since this aerosol may be more susceptible to aging we examine its features more closely. Backtrajectories from the profile at 12.3°S, 11°E show the aerosols are coming from similar source regions (Fig. 12d), and become distributed vertically primarily by variations in the advection speed. One main aerosol layer is centered on 5 km, aged ~4 days since emission, and a slightly older smoke

layer of $\sim$5 days in age is centered on 3 km (Fig. 12b,c). The younger aerosol aloft is connected to stronger upper-altitude winds also transporting moisture (Fig. 12a), consistent with climatological expectations (Adebiyi et al., 2015; Adebiyi and Zuidema, 2016; Pistone et al., 2021). These generate relative humidities exceeding 80% above 4 km when combined with the cooler high-altitude temperatures (Fig. 12a). Although there are two main aerosol plumes, the potential temperature profile is
385 of a thermally stratified atmosphere containing many thinner seemingly well-mixed layer separated by discrete stability jumps (Fig. 12c). The water vapor mixing ratio profile (Fig. 12c) indicates there is only one truly well-mixed layer, capping the upper aerosol plume between 5.3-5.8 km with slackening winds. The lack of vertical mixing indicates the smoke plume heights are likely set above land. The upper-level aerosol plume registered both the highest OA:BC mass ratio and the highest SSA of the ORACLES-2016 campaign. More intense fires, with lower OA:BC are typically able to reach higher altitudes (Martin et al.,
2010), but the higher OA:BC and $NO_3$:BC mass ratios aloft (Fig. 12b) may instead indicate more condensation of the emitted vapors aloft, aided by cooler temperatures and higher relative humidities (Li et al., 2018).

Secondary aerosol formation is expected to proceed more quickly when $\frac{BC}{\Delta CO}$ ratios are lower (Vakkari et al., 2018), because the precursor gases needed for nucleation may be more available (Yokelson et al., 2009). We first confirm that the flight's $\frac{BC}{\Delta CO}$ values remain statistically-similar as a function of *f44*: these remain within 7.5-7.9 $*10^{-3}$ independent of *f44* (Fig. 13a; see
Figs. S10-S11 for the same analysis for the other flights). We interpret this to mean that the aerosol are emitted from similar sources over a two-day time span, with no wet deposition throughout. The corresponding OA:BC mass ratio decreases from 14.2 to 9.8 (Fig. 13b) - an approximate 35% decline in OA:BC over a span of 1-2 days. The corresponding SSA reduces from 0.89 to 0.865. The mean $AAE_{470-660}$ decreases from 1.25 to 1.21 ($\pm$0.07-0.08) as a function of the three *f44* bins, a statistically insignificant decline.

An evaluation of the changes to the particle size distribution with *f44*, normalized with respect to BC as a control for dilution, indicates the processes of condensation and coagulation - and volume loss consistent with the mass loss. As the aerosol ages chemically, the LDMA median diameter increases from $\sim$ 170 nm to $\sim$ 205 nm (Fig. 14a), mostly because the number of particles with diameters < 100 nm declines. BC particles are typically larger than OA particles (e.g., Fig. S6), and the reduction in the number of small particles indicates coagulation of the OA particles upon the larger BC particles. It also
indicates that most of the vapors are condensing on the larger BC particles, as opposed to forming new particles by nucleation. The total LDMA and CPC particle number concentrations reduce from approximately 1200 cm$^{-3}$ to 500 cm$^{-3}$ with respect to BC, and 2400 cm$^{-3}$ to 1500 cm$^{-3}$, respectively (Fig. 14c). The large difference in the two number concentrations likely reflects an instrument difference; both instruments agree there is no net production of the smaller particles. The combined effect of condensation and coagulation results in an increase in the fraction of BC-containing particles from 0.18, to 0.23 and then
0.27 as *f44* increases. The evolution towards larger sizes would increase the SSA, all else equal. Instead, the SSA decreases in response to the decrease in OA, indicating again that changes in particle size do not dominate the SSA changes.

At the same time, the LDMA-determined particle volume decreases (Fig. 14b), indicating genuine particle mass loss that is consistent with the decrease in OA:BC. One mechanism for the mass loss could be evaporation through dilution. The selection for data samples with OA > 20 $\mu$g m$^{-3}$ focuses the analysis on the aerosol plume center, and a normalization by BC provides
an additional control, leading us to discount this mechanism. Aqueous phase reactions and mid-level cloud processing could

potentially also contribute to the oxidation increase and loss of free-tropospheric OA mass concentration. This is partially controlled for by only selecting free-tropospheric data samples with RH<80%. Mid-level clouds, produced by dry convection saturating the top of the land boundary layer, can occur (Adebiyi et al., 2020), but are not a dominant presence on this day or other ORACLES flight days. This suggests to us that the reduction of free-tropospheric OA through aqueous phase reactions is of secondary importance (becoming even more so with distance from the continent).

Instead, we speculate the dominant loss mechanism can be increasing oxidation through fragmentation, which can release higher-volatility particles that can then be subsequently removed (e.g., Jimenez et al., 2009). Figs. 14c-d support that interpretation: both the LDMA and CPC total particle number concentrations decrease with *f44*, consistent with processes occurring at the surface of the larger particles - either coagulation or surface reactions. The oxidative environment, inferred from $\frac{O_3}{\Delta CO}$, remains constant with *f44*, but these measures for oxidation may have reached their upper limit. The reduction in the total non-BC aerosol mass, which reflects a reduction in the combined $OA+NH_4+NO_3$ mass but not $SO_4$ is nevertheless in concert with the OA:BC decrease. The constancy of $SO_4$:BC with *f44* (not shown) confirms the aerosol is aged, as the lifetime of $SO_2$ is 1-2 days, after which its conversion to $SO_4$ will have ended. In summary we interpret Fig. 14 to reflect changes in the particle size distribution induced by condensation, coagulation and mass loss through fragmentation .

The backtrajectories do not show lower-level westerlies, in contrast to the *in-situ* profiles. We conjecture that the daytime aircraft sampling sampled a land breeze below 4 km that is converging above a warming continental surface and is not represented in the GFS meteorology. The ERA5 dataset, which has an hourly temporal resolution, might be able to address this hypothesis. Not shown, the marine boundary layer top is at one km, and the boundary layer did not include any BC, consistent with a slow entrainment time scale for aerosols (Diamond et al., 2018).

## 6   Does comparing to aerosol measured at Ascension Island indicate ongoing compositional changes?

A third possible mechanism for the loss of the overall particle mass could be through thermodynamics, consistent with the increase in $NO_3$:BC with altitude (Fig. 12b). A comparison to the aerosol properties measured at the more remote location of Ascension island by CLARIFY (Table 3) supports the speculation that fragmentation of oxidized aerosol may be contributing to mass loss, by ultimately releasing some aerosol that can evaporate through photochemistry, similar to the younger aerosol sampled on 9/24/2016. Table 3 compares values for the aerosol parameters derived from the six ORACLES flights to the free-tropospheric values reported within Wu et al. (2020) and Taylor et al. (2020). The ORACLES 8/31/2017 flight coincided with CLARIFY and occurred halfway to the island (Fig. 1). Important to this comparison, the ORACLES and CLARIFY $\frac{OA}{\Delta CO}$, BC and SSA values compared well on the intercomparison flight held on 18 August 2017 (Barrett et al., 2022). Their similar $\frac{BC}{\Delta CO}$ ratios (Table 2) also equal the maximum values inferred from the surface-based measurements at Ascension between June to October (Che et al., 2022).

Overall, CLARIFY sampled more BC-enriched particles in the free troposphere above Ascension in both number and mass, with slightly lower SSA, higher $MAC_{660}$s, and larger particle sizes. The OA:BC mass ratios are lower, mostly because the OA masses themselves are much lower, typically $< 4$ $\mu$g m$^{-3}$ (Wu et al., 2020). Interestingly, September African BBA

sampled near the Amazon Tall Tower Observatory indicated BC mass and number fractions on par with CLARIFY, with OA:BC mass ratios between 2.5-5.0, although the sampling is only from one aircraft flight (Holanda et al., 2020). The H:C, O:C and OA:OC CLARIFY values (1.2-1.4, 0.7-0.8, 2.3) indicate slightly more oxidized aerosol at Ascension (Wu et al., 2020) than for ORACLES, with the exception of 31 August 2017 (Fig. 8). ORACLES 8/31/2017 flight values tend to be intermediate to the CLARIFY and ORACLES-2016 numbers.

Other prior field campaigns have highlighted a small net OA loss as BBA ages beyond a day (e.g., Capes et al., 2008; Jolleys et al., 2012, 2015; Hodzic et al., 2015; Konovalov et al., 2019; Farley et al., 2022), attributed to evaporation through dilution in Jolleys et al. (2012). The extent of the OA reduction reported here - a factor of at least two between ORACLES and CLARIFY OA masses - suggests a different process must be dominant. Nevertheless, we examine if dilution could be factor, through comparing the number and volume size distributions measured by the TDMA during the 8/31/2017 flight at STP and at 150°C (Fig. S7). The heating is used as a proxy for dilution because processes respond to aerosol volatility. Although the TDMA size distribution does not extend beyond 0.2 $\mu$m at most, missing the bulk of the aerosol mass, the size distributions change little between the temperatures, supporting the inference that evaporation through dilution does not explain differences in OA:BC between the two campaigns.

Another significant difference between ORACLES-2016 and CLARIFY is the inorganic nitrate (IN) fraction. Nitrate only contributes 10% to the total aerosol mass analyzed here, and only 8% of the total free-tropospheric aerosol mass during CLARIFY (Wu et al., 2020). The fraction that is inorganic nitrate is even smaller. Interestingly, the 8/31/2017 inorganic nitrate fraction was 51% - intermediate to the ORACLES-2016 and CLARIFY values (Table 3). This suggests that organic nitrate may become converted to inorganic nitrate as the aerosol ages. The air sampled during ORACLES was mildly acidic (Fig. S10), based on a simplification of the $NH_{4,measured}/NH_{4,predicted}$ relationship put forth in Zhang et al. (2007). A mild acidity will slow the rate of inorganic acid formation, and may help explain the lower IN fraction for ORACLES (20%-25%). Inorganic ammonium nitrate is held responsible for an increase in SSA with height at Ascension (Wu et al., 2020), as thermodynamic partitioning favors the particle phase at higher altitudes. The nitrate fraction is never large, and the coating on the BC will be dominated by OA by mass, so that the IN fraction may be more valuable as an indicator of ongoing oxidation that is also capable of increasing the MAC (Shrivastava et al., 2017). The sulfate fraction is similar between the two campaigns, including the 8/31/2017 flight, and seems unlikely to explain the differences in the MAC between the campaigns. Increasing coagulation of smaller particles upon the BC particles could help explain why the particle diameters are larger at Ascension compared to ORACLES-2016, at the same time that evaporation through photochemistry increases the fraction of BC-containing particles while decreasing the overall OA:BC mass ratio. It is also possible that the non-Lagrangian sampling is introducing a bias. A Lagrangian analysis of filter samples did find increased aerosol volatility and continued OA loss in ORACLES-CLARIFY resampled aerosol (Dang et al., 2022).

We also examine if a portion of the OA can thermodynamically repartition. We composite OA:BC, NO$_3$:BC and aerosol age by the free-tropospheric RH for all six flights to illuminate how much thermodynamical partitioning by altitude may be occurring. The higher altitudes in the free troposphere are also often more humid (e.g., Fig 12, shown more statistically in Adebiyi et al. (2015) and Pistone et al. (2021)). Physically-younger aerosol is more likely to occupy a more (relatively) humid,

colder free troposphere at higher altitudes than is older aerosol (e.g., Fig. 12). The mean $NO_3$:BC ratio decreases by almost 50% as the free troposphere RH decreases from 70% to 30% (Fig. 15a), consistent with a thermodynamic repartitioning to the gas phase. The mean OA:BC mass ratio only reduces from $10.5 \pm 2.6$ for RH = 60-80% to $9.9 \pm 2.1$ for 20% < RH < 40%, a statistically insignificant decrease. A thermodynamical repartition can only explain a relative change in OA:BC of less than 10%, if that. The small change in OA:BC, if real, may also reflect moisture contributing to the OA mass loss through increasing OH uptake and/or fragmentation (Li et al., 2018), or because at higher altitudes, less-oxidized material is continuing to condense upon the pre-existing organics, ultimately favoring the evaporation of OA into the gas phase.

## 7  Discussion

This study extends and strengthens an earlier analysis begun within Dobracki et al. (2022). One flight, on 9/24/2016, has consistent BC:CO ratios as a function of the *f44* chemical aging marker, interpreted to mean emissions from similar sources over a few days with no wet deposition. At the lower altitudes with weaker offshore winds, condensation and coagulation explain an increased particle size for the slightly older aerosol. Increasing oxidation through fragmentation, which can then release higher-volatility particles through photochemistry, can explain the reduced overall mass. Dilution is not considered as influential as for northern hemisphere boreal fires, because the smoke distribution is so broad and loadings so large, composed of many small continental small fires that become homogeneized before the smoke is advected offshore. An increase in SSA with height is primarily explained by more OA aloft. This contrasts with the increase in SSA with height at Ascension attributed to an increase in ammonium nitrate (Wu et al., 2020), which may reflect changes in the aerosol composition occurring with further transport offshore. Level-leg measurements from six flights demonstrate how optical properties relate to chemical and physical composition and can be compared to values from Wu et al. (2020) and Taylor et al. (2020) made at Ascension Island. Further focus on data from the 8/31/2017 flight helps connect interpretations of ORACLES versus CLARIFY aerosol characteristics.

The total aerosol concentration exceeds the total SP2-derived BC number by a factor of 2.5 to 7, from which we infer that at least one-half of the non-BC aerosol remains externally mixed with the BC. The BC itself, because of its transport within multiple days within broad, dense smoke plumes, is most likely internally mixed with other aerosols, confirmed by independent electron microscopy measurements (Dang et al., 2022). Because the BBA is at least 4 days old, and as already shown within Taylor et al. (2020) and Denjean et al. (2020b), the BC can be treated as compacted. Taylor et al. (2020) find a better fit to the CLARIFY MAC measurements using the semi-empirical models of Liu et al. (2017) and Chakrabarty and Heinson (2018) than to a core-shell Mie model, but Lee et al. (2022) conclude a core-shell mode can be successfully applied once particle-by-particle differences are accounted for. The 2016 data from ORACLES lacks SP2 mixing state date with which to better evaluate optical fits, but an independent assessment could be pursued using the SP2 coating-resolved ORACLES data from 2017 and 2018 (Sedlacek et al., 2022).

A highlight of this study is its $SSA_{530}$ best-fit regression on OA:BC: $SSA_{530}=0.801+0.0055*(OA:BC)$. The range of OA:BC of 7 through 14 equates to an SSA variability of 0.83 to 0.89. This provides a straightforward connection between the BBA

chemical and optical properties that is useful for the modeling of the direct aerosol radiative effect. Of course, use of such a best-fit is only effective if the model OA:BC mass ratios are realistic. Given that OA:BC mass ratios are often too low in models, their absorption of sunlight will also be overestimated (Brown et al., 2021). This study adds to literature indicating that OA model estimates made by multiplying the organic carbon by a factor of 1.4 will underestimate OA in this (and other) regions (Aiken et al., 2008; Tsigaridis et al., 2014; Shinozuka et al., 2020; Doherty et al., 2022). This study's OA:OC mass ratios of $2.2 \pm 0.1$ is also shown for the Atomic Tomography mission (Hodzic et al., 2020).

More sophisticated aerosol schemes can, in contrast, overestimate OA:BC mass ratios over the southeast Atlantic (Chylek et al., 2019), suggesting the loss of OA with aging or slower SOA production processes (Kroll and Seinfeld, 2008; McFiggans et al., 2019) may also be under-accounted for. For the southeast Atlantic region, far removed from urban and industrial sources of pollution, continued production of SOA after 1-2 days is expected to remain minor (e.g., O'Brien and Kroll, 2019). This contrasts with northern hemisphere fire emission sources. Brown carbon production has been linked to low OA:BC ratios (Saleh et al., 2014; McClure et al., 2020), but this does not seem supported by the admittedly-limited ORACLES measurements of AAE and MAC, perhaps because brown carbon is typically more closely linked to primary than to secondary organic aerosol. Many prior studies find continuing oxidation of OA (see review by Shrivastava et al., 2017). This will be more important for remote environments containing thick smoke layers lacking additional pollution sources producing the precursor gases for additional SOA. Further work is needed to better support the process conclusion of this study, named that evaporation through fragmentation/photochemistry is the dominant chemical aging process over the southeast Atlantic, but nevertheless this study indicates the importance of developing sophisticated SOA schemes (e.g., Lou et al., 2020) for this regional climate.

September is unique in that meteorology and fire processes conspire to accentuate the direct radiative warming of the southeast Atlantic. August is likely the month with the most fires in southern Africa (Scholes et al., 1996), but the upper-level winds that transport the aerosol depend on a strong heat low over southern Africa, and don't become well-established until September (Adebiyi and Zuidema, 2016; Kuete et al., 2020). The winds occur to the north of the heat low, and only dry convection lofts the aerosols to their altitude. The winds distribute aerosol as far away as south America (Holanda et al., 2020), so that the entire south Atlantic is covered by a blanket of highly-absorptive aerosol, with submicron BC mass fractions far exceeding the 2-10% estimated for western north America (Garofalo et al., 2019). The strong September upper-level winds also discourage subsidence (Chaboureau et al., 2022), and the cloud cover and height are also affected directly by meteorology during this month (Adebiyi and Zuidema, 2018; Zhang and Zuidema, 2021). Less of the aerosol reaches the cloud top, reducing aerosol entrainment into the cloud layer (Zuidema et al., 2018; Shinozuka et al., 2020; Doherty et al., 2022) and ability to influence the cloud top inversion strength (Herbert et al., 2020). The net radiative impact will primarily be the direct aerosol radiative effect of the aerosol aloft then, lending further weight on model representation of SSA (Mallet et al., 2021). The remaining ORACLES data from 2017 and 2017, for which SP2 mixing state data are available, will be used to support further investigation of SSA-aerosol composition relationships in a follow-up study.

*Data availability.* The data are available through doi=10.5067/Suborbital/ORACLES/P3/2016_V2 and doi=10.5067/Suborbital/ORACLES/P3/2017_V2.

*Author contributions.*  The present work was conceived by PZ, AD, PS and SH. SF contributed to the HiGEAR data analysis, AS provided the BC datasets and PS the WRF-AAM model age estimates. Portions of this work first appeared in the M.S. thesis of A.D at U. of Hawaii. PZ led the writing with AD providing most of the figures. All authors contributed to the final writing.

*Competing interests.*  PZ is a guest editor for the ACP Special Issue: "ACP special issue: New observations and related modelling studies of the aerosol–cloud–climate system in the Southeast Atlantic and southern Africa regions" .The other authors declare no competing interests.

*Acknowledgements.*  ORACLES is a NASA Earth Venture Suborbital-2 investigation, funded by the US National Aeronautics and Space Administration (NASA)'s Earth Sciences Division and managed through the Earth System Science Pathfinder Program Office (grant no. NNH13ZDA001N-EVS2). This work was further supported by the US Department of Energy grant DE-SC0018272 to PZ and PS and DE-SC0021250 to PZ. We thank Hugh Coe, Huihiu Wu and Jonathan Taylor for interesting initial conversations that in particular encouraged us to examine the inorganic nitrate fraction.

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

**Table 1.** Instrument Table

| Measurement | Instrument (Manufacturer) | Notes |
|---|---|---|
| OA, NO$_3$, SO$_4$, NH$_4$ masses, *f44, f60*, O,C,H,OC | HR-ToF-AMS (Aerodyne Inc.) | chloride excluded |
| BC mass, number | SP2 (DMT) | operated by HiGEAR in 2016, A. Sedlacek in 2017 |
| particle size distribution | LDMA (TSI 3071A) | 10-550 nm mobility diameter |
| particle number concentration | CPC (TSI 3010) | diameter > 10nm |
| aerosol absorption (470, 530, 660 nm) | PSAP (Radiance Research) | Virkkula (2010) wavelength-averaged correction |
| aerosol scattering (450, 550, 700 nm) | nephelometer (TSI 3563) | Anderson and Ogren (1998) correction |
| CO, CO$_2$, O$_3$ | Los Gatos Research | |

abbreviations provided in text

**Table 2.** Comparison to other published $\frac{BC}{\Delta CO}$ ratios

| Fuel/Geographic Source | $\frac{BC}{\Delta CO}*10^{-3}$ | reference |
|---|---|---|
| savannah | 2-15 | Vakkari et al. (2018) |
| grass | 10-17 | Vakkari et al. (2018) |
| savannah | 7.9 | Andreae (2019) |
| agriculture | 5.6 | Andreae (2019) |
| savannah | 5.9 | Akagi et al. (2011) |
| agriculture (crop residue) | 7.4 | Akagi et al. (2011) |
| NW African agriculture, smouldering | 7.2 | Capes et al. (2008) |
| southern Africa (SAFARI) | 7.0 | Formenti et al. (2003) |
| Ascension Island, August | 8.7-13.4 | Wu et al. (2020) |
| **this study** | **9.6** | |

all $\frac{BC}{\Delta CO}$ values are dimensionless. Methods for deriving the BC mass concentration may vary between the studies. Most CMIP6 models rely on the Akagi et al. (2011) emission factors.

**Table 3.** Comparison of level-leg mean values to CLARIFY

| | CLARIFY | September 2016 | 31 August 2017 |
|---|---|---|---|
| BC mass frac. (%) | 13-15 | 5.4-9.2 | 7 |
| BC num. frac. (%) | $39 \pm 7$ | 15-40 | 30-35 |
| SSA$_{530}$ | $\sim 0.84$ | 0.85-0.88 | 0.83-0.86 |
| MAC$_{BC,660}$ (m$^2$ g$^{-1}$) | 11-12 | 9.5-11.5 | 10-11.5 |
| OA:BC mass | 4-5 | $10\text{-}14 \pm 2$ | 8-10 |
| LDMA median diam. (nm) | 232 | 140-180 | 180-200 |
| BC core diam. (nm) | | 130-150 | 150-160 |
| IN frac. (%) | 100 | $\sim 25$ | $\sim 50$ |
| *f44* | 0.19-0.22 | 0.18-0.22 | 0.215 |
| $\frac{OA}{\Delta CO}*10^{-2}$ | 4.2-6.4 | 6-11 | 6.5-11.5 |

CLARIFY free-tropospheric values taken from Wu et al. (2020) and Taylor et al. (2020), based on CLARIFY 033-039 and 045-051 flights. CLARIFY BC number fraction calculated relative to PCASP-derived total number concentrations.

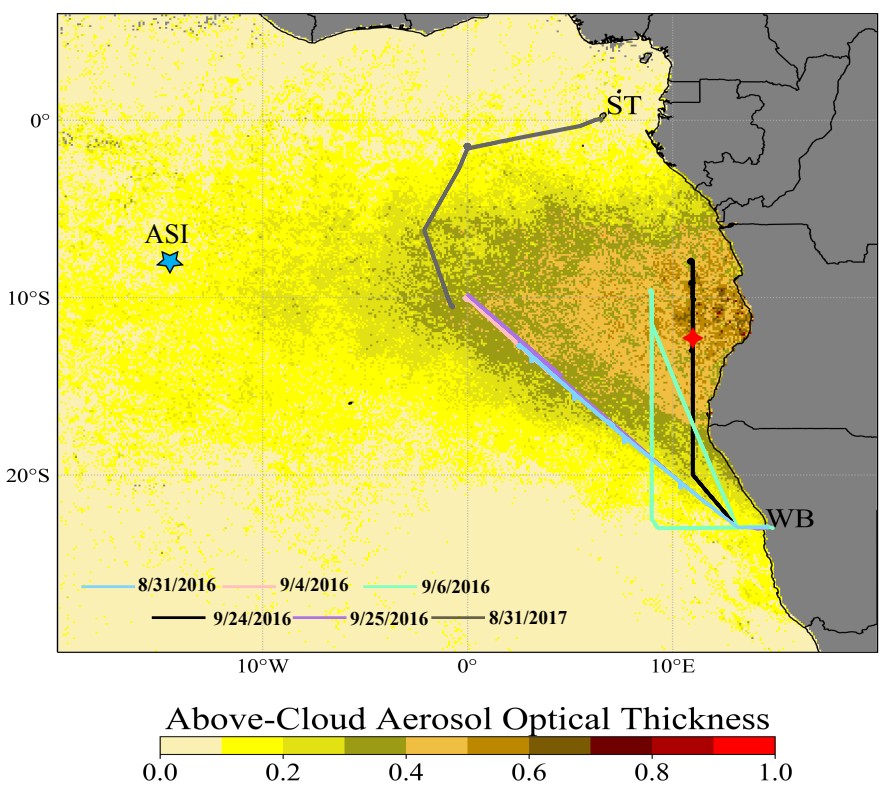

**Figure 1.** Terra MODIS Above Cloud Aerosol Optical Depth (Meyer, 2015) for September 2016 overlaid with the tracks of the 6 flights selected for this study. The location of the profile shown in Fig. 12 is indicated with red diamonds. ST=Sao Tome; WB=Walvis Bay; ASI=Ascension Island.

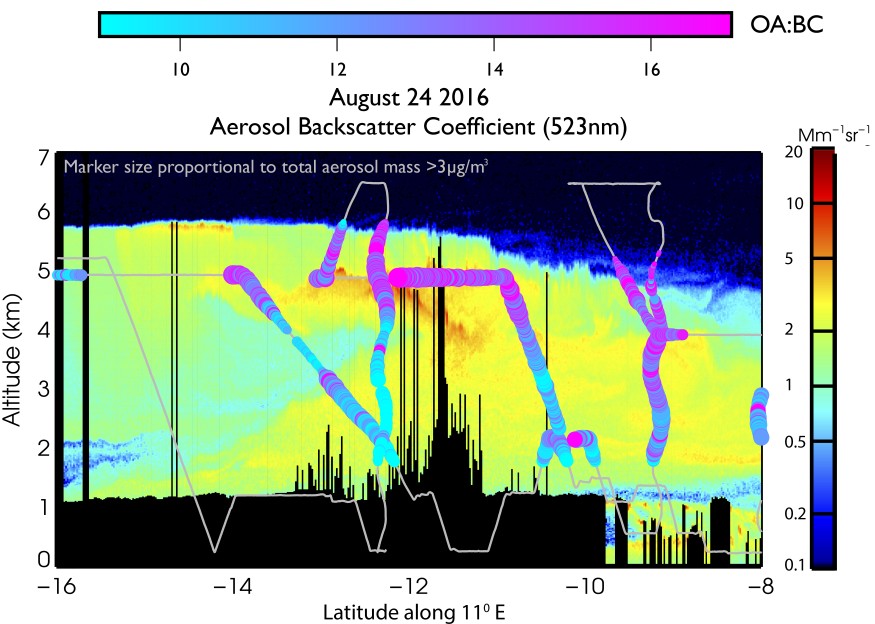

**Figure 2.** 24 September 2016 flight track with colorized OA:BC mass ratios superimposed on High Spectral Resolution Lidar-2 523 nm aerosol backscatter imagery collected along 11°E by the overflying ER-2 plane, near in time to the P-3 plane location's at 10°S.

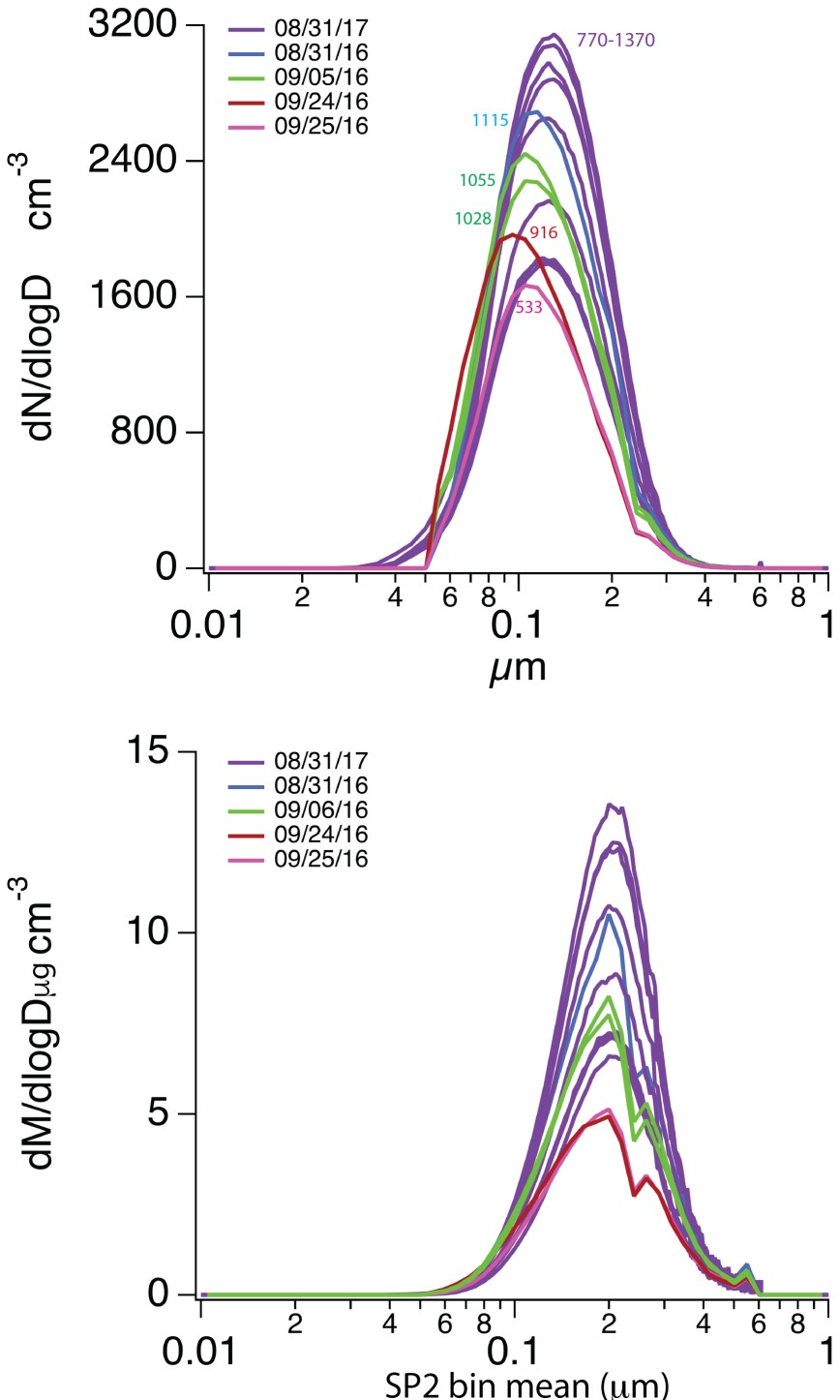

**Figure 3.** Leg-averaged SP2 size by number (top) and mass (bottom) distributions for the level legs detailed in Table S2. Total leg-averaged SP2 number concentrations included in top plot. The kink at 2.5 $\mu$m mass-mean diameter for the 2016 data is an instrument artifact.

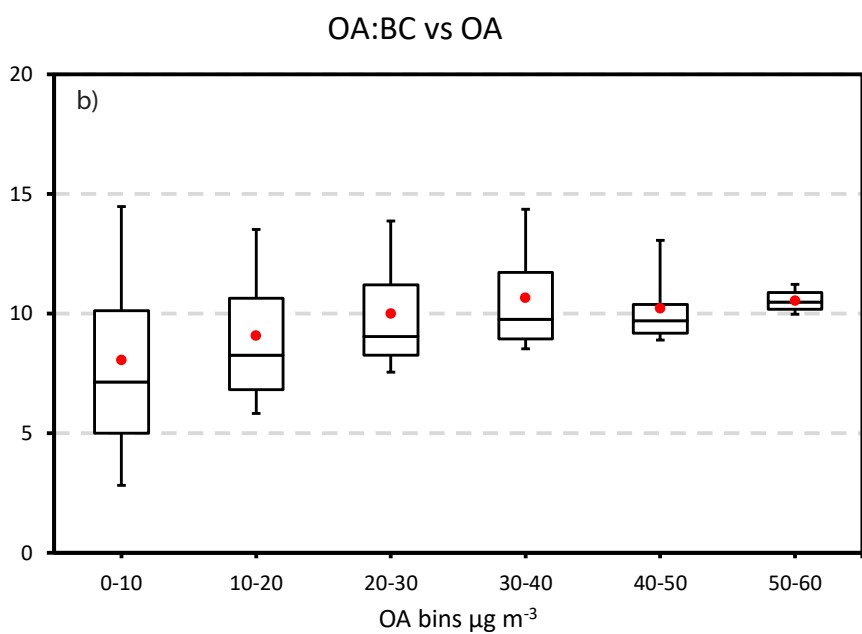

**Figure 4.** OA:BC mass ratio, shown using 10, 25, 50, 75 and 90th percentiles with means in red, composited by OA mass.

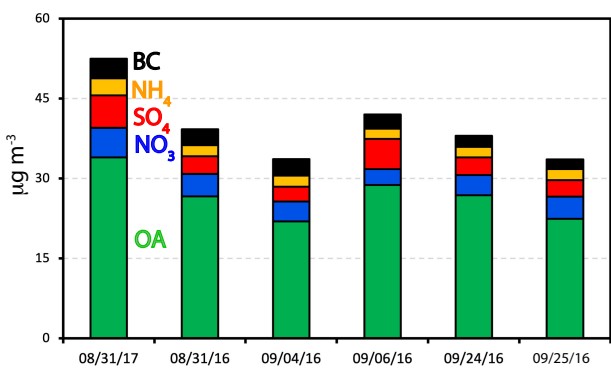

**Figure 5.** Distribution of the bulk chemical species masses for each flight, for five-second samples with OA>20 $\mu$g m$^{-3}$.

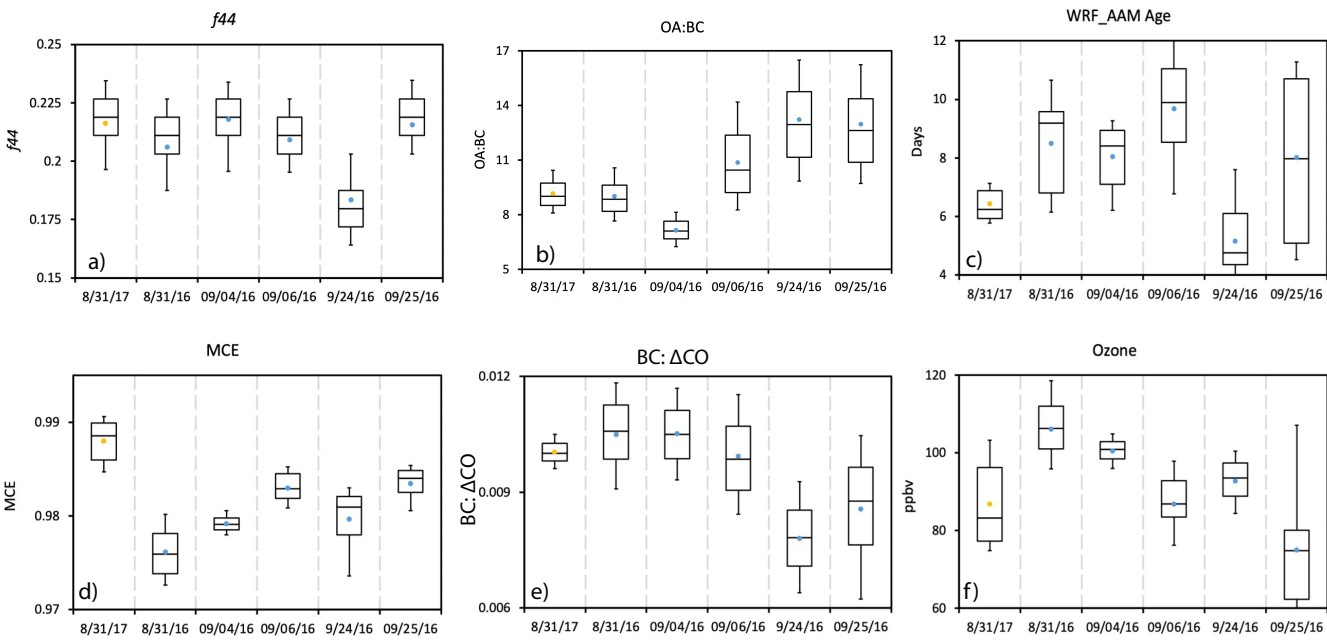

**Figure 6.** a) *f44*, b) OA:BC mass ratio, c) model-derived time since emission (age), d) MCE, e) $\frac{BC}{\Delta CO}$ as a non-dimensionalized ratio, and f) ozone, all for each indicated flight. Whiskers represent the 10th and 90th percentiles, boxes illustrate the 75th and 25th percentiles with a line indicating the median and yellow (2017) and blue (2016) filled circles representing the mean. OA>20 $\mu$g m$^{-3}$ only.

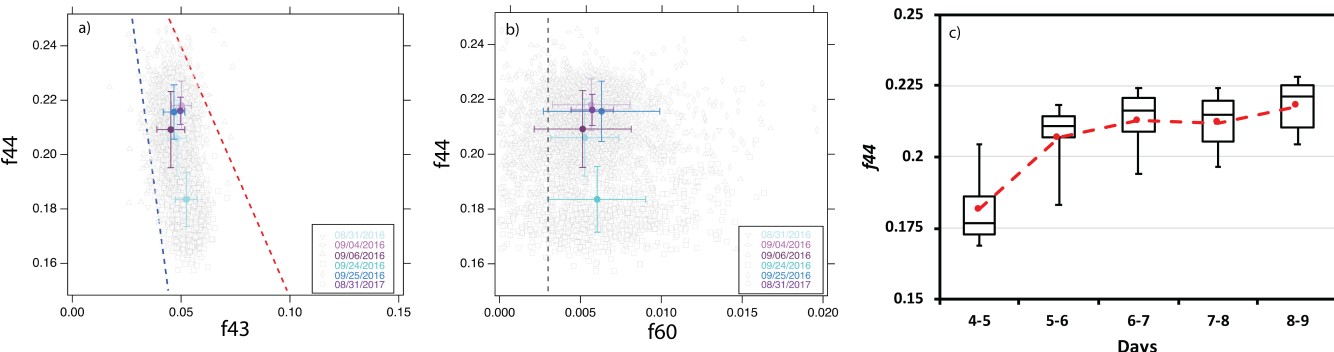

**Figure 7.** a) *f44* versus *f43* for the six flights where OA > 20 $\mu$g m$^{-3}$. Averages ($\pm$ standard deviation) are colored by flight date, grey boxes indicate individual data points. b) similar to a), for *f44* vs *f60*. Blue and red dashed lines define the parameters for ambient oxygenated OA, following Ng et al. (2010). c) *f44* versus the model-derived physical age for the six flights combined.

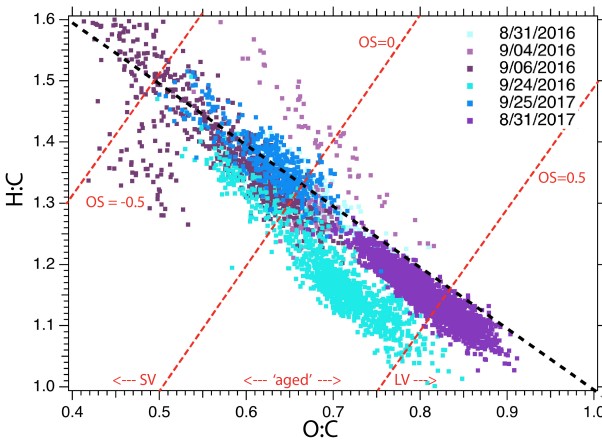

**Figure 8.** Hydrogen to carbon (H:C) molar ratio versus oxygen to carbon (O:C) molar ratio, colored by flight date, shown at the native 5-second time resolution. Superimposed are lines of constant oxidation state (OS, defined as 2* O:C - H:C; Kroll et al., 2011), used to define semi-volatile (SV), aged and low-volatile (LV) oxidized organic aerosol (OOA) regimes.

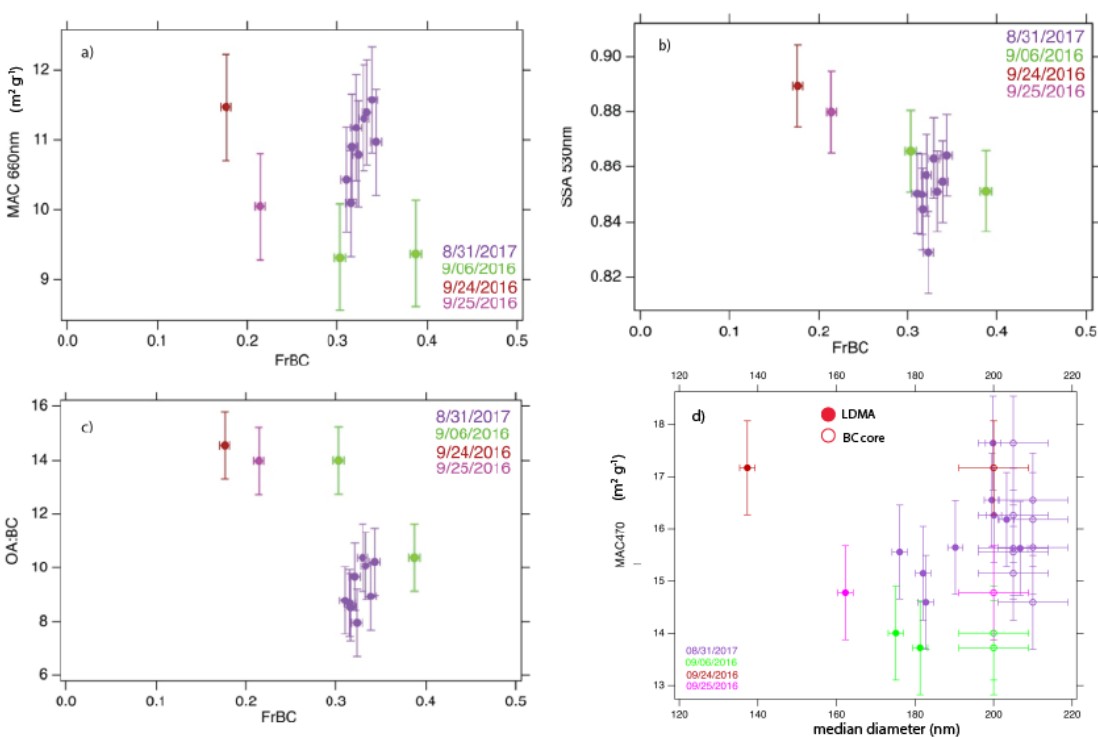

**Figure 9.** a) Mass absorption coefficient ($MAC_{BC,660}$; units of $m^2\ g^{-1}$) at 660 nm wavelength versus the fraction of black-carbon-containing particles (FrBC), colored by flight day. b) same as a) but for $SSA_{530}$ versus FrBC. c) same as a) but for OA:BC mass versus FrBC. d) $MAC_{BC+OA,470}$ ($m^2\ g^{-1}$) versus LDMA median and mass-median BC core diameter. All for the level legs listed in Table S2.

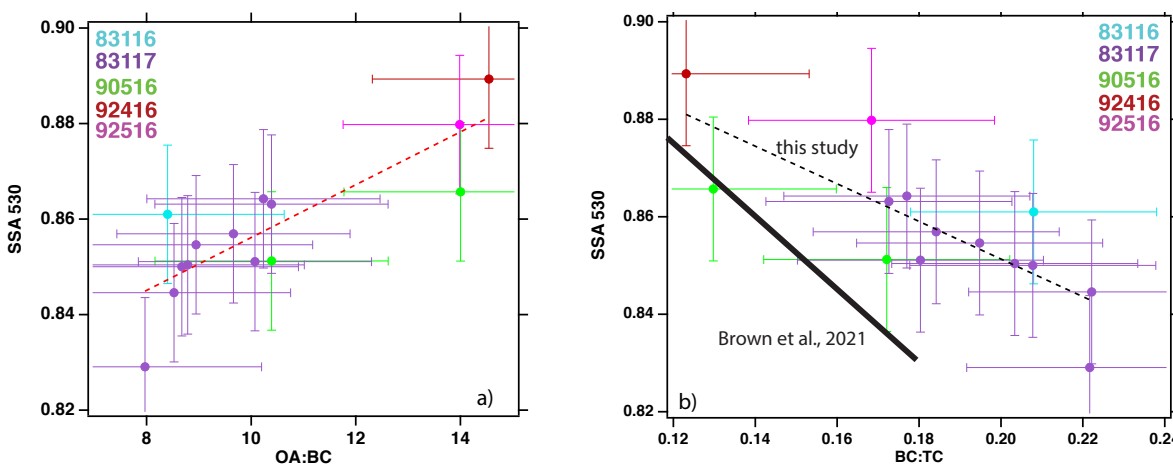

**Figure 10.** a) Level-leg-mean $\pm$ standard deviation values for $SSA_{530nm}$ versus the OA:BC mass ratio, colored by flight. The best-fit line is represented by SSA=0.801+0.0055*(OA:BC) ($r$=0.84). b) same as a) but for $SSA_{530nm}$ versus the BC:TC mass ratio, where TC=BC+organic carbon. The best-fit line is SSA= 0.93-0.39*(BC:TC), ($r$= -0.79). Times and spatial ranges of the level-legs provided in Table S1. Also shown is the SSA parameterization put forth within Brown et al. (2021), namely $SSA_{550nm}$=0.969-0.779*(BC:TC).

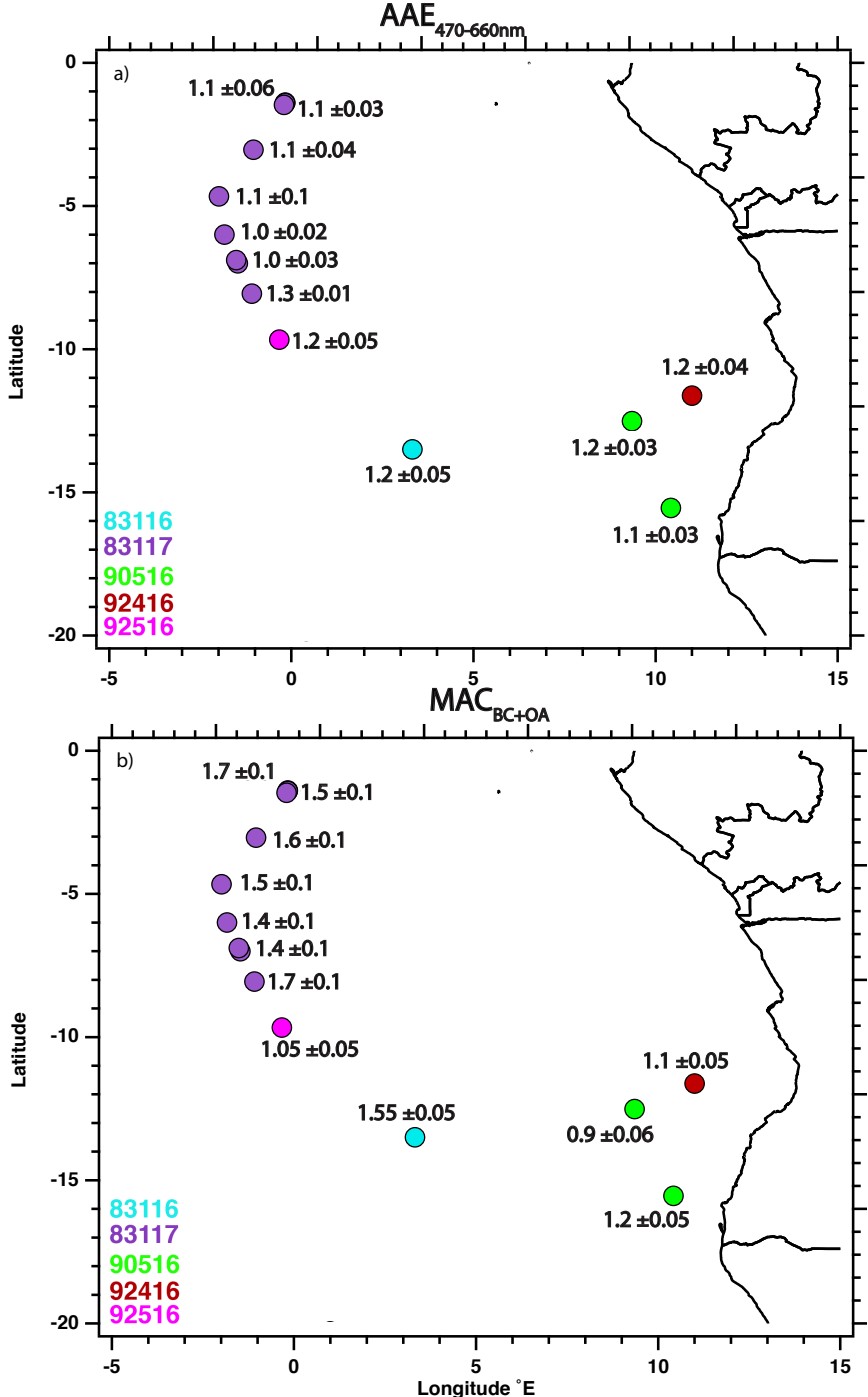

**Figure 11.** a) Absorption Ångström exponent (470-660 nm), b) $MAC_{OA+BC,470}$ ($m^2/(^{-1})$), for the same level legs shown in Figs. 9-10, similarly colored by flight date.

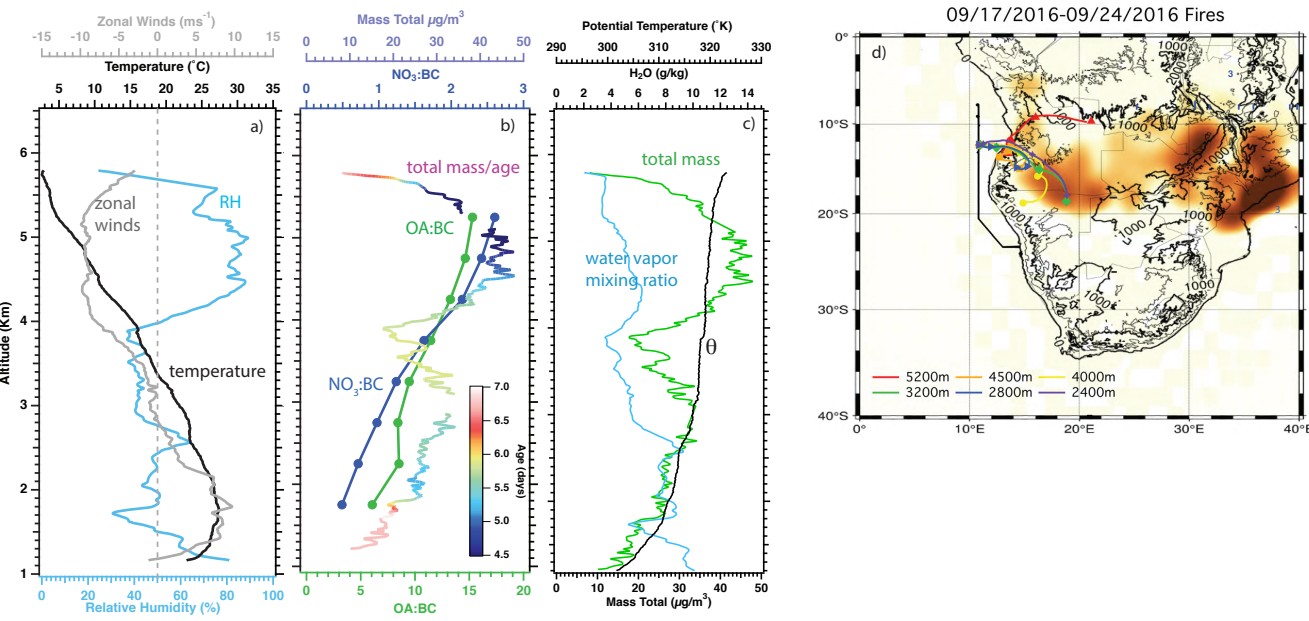

**Figure 12.** 24 September, 2016 (12.34°S, 11°E) vertical profiles of a) relative humidity (%; blue), zonal winds (m s$^{-1}$; grey) and temperature (°C), and b) organic aerosol to black carbon mass ratio (OA:BC; green), total nitrate to black carbon ratio (NO$_3$:BC; blue) averaged every 500 m (approximately 2 minutes of data), and c) total mass concentration (OA + BC + SO$_4$ + NO$_3$ + NH$_4$ in $\mu$g m$^{-3}$; 1Hz resolution) colored by aerosol age. d) HYSPLIT trajectories, initialized on 12 UTC of 24 September, 2016 with markers placed at 0 UTC of preceeding days, superimposed on map of fires detected between 9/17/2016-9/24/2016.

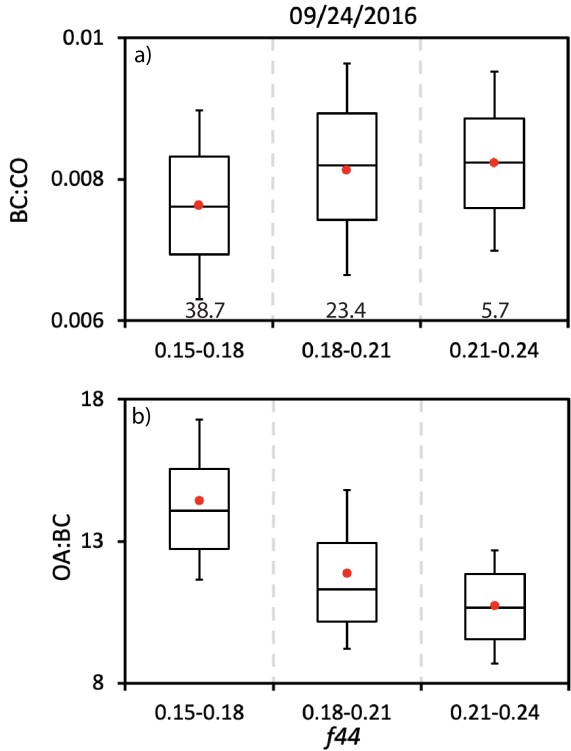

**Figure 13.** a) $\frac{BC}{\Delta CO}$ ratios (dimensionless) as a function of *f44* for the 9/24/2016 flight. The number of minutes contributing to each *f44* bin is stated at bottom of panel. b) same for OA:BC. Whiskers represent the 10th and 90th percentiles, boxes illustrate the 75th and 25th percentiles with a line indicating the median and a red filled circle the mean. OA>20 $\mu$g m$^{-3}$ only.

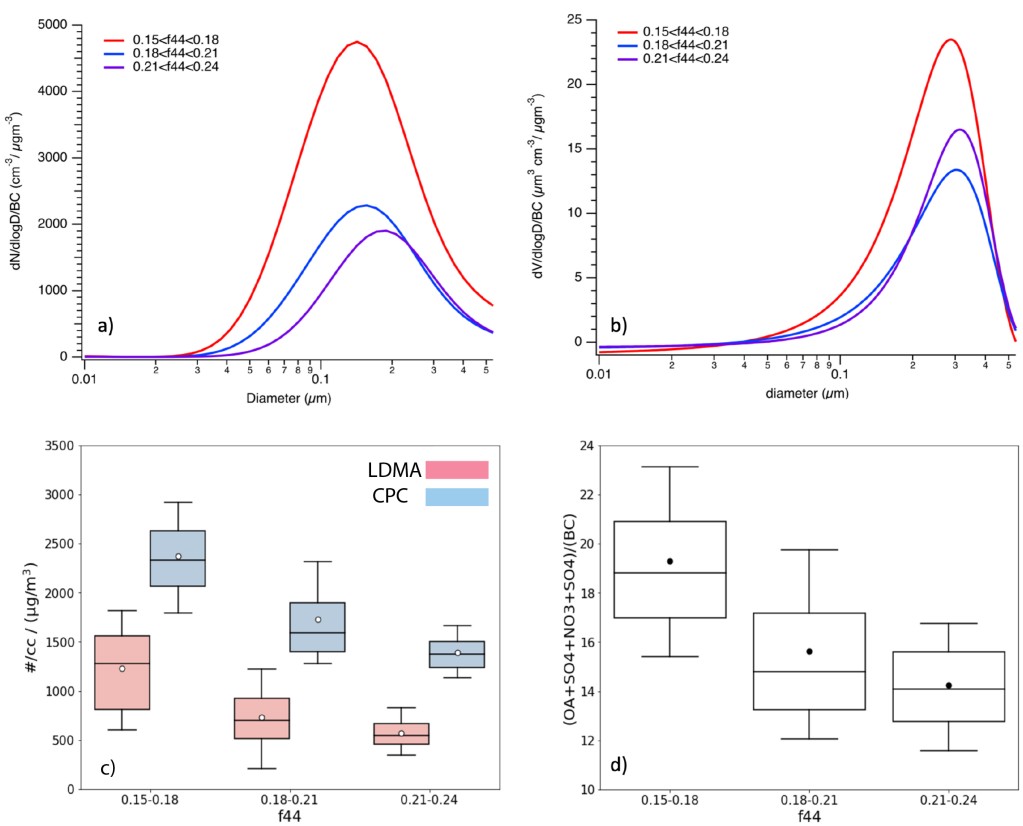

**Figure 14.** a) 24 September, 2016 LDMA-derived number particle size distribution for three *f44* bins (0.15-0.18 - red; 0.18-0.21-blue; 0.21-0.24 - purple), divided by the BC mass. b) same as a) but for the LDMA-derived particle size distribution. c) LDMA and CPC particle number concentration as a function of *f44* and d) ratio of non-BC total mass (OA + $SO_4$ + $NO_3$ + $NH_4$) to BC, as a function of the three *f44* bins. All data are selected from OA >20 $\mu$g m$^{-3}$.

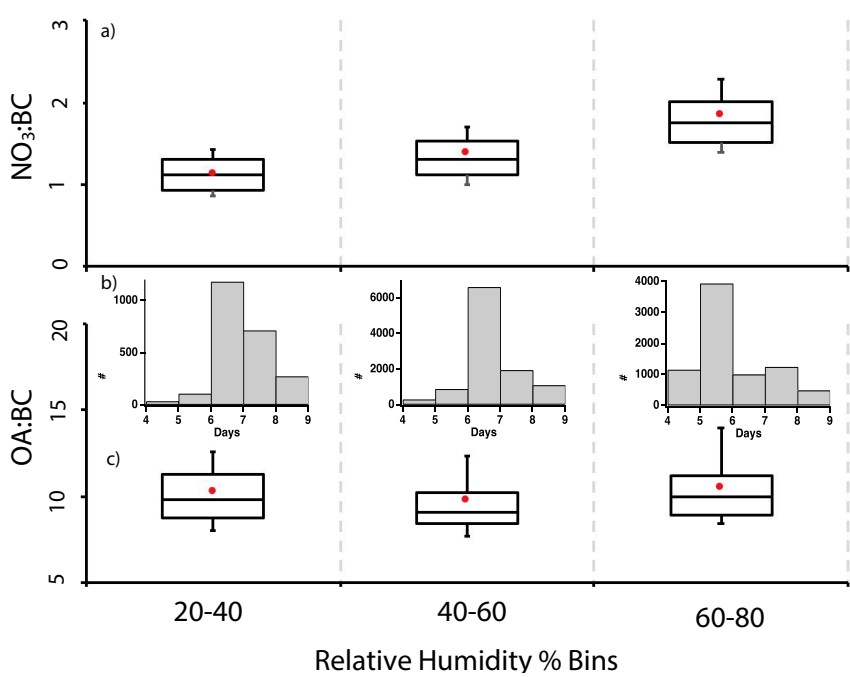

**Figure 15.** a) NO$_3$:BC and c) OA:BC mass ratios for the 6 selected flights as a function of relative humidity, for OA>20 $\mu$g m$^{-3}$ at STP. The 10th, 25th, median, 75th and 90th percentiles are indicated using box-whiskers, the mean with solid red circle and marker. b) corresponding distribution of aerosol ages within each relative humidity range, with the y-axis indicating the number of 1-sec samples.

## 1 Flight details

Aerosol forecast maps indicate the spatial sampling of the aerosol plumes for 31 August, 4 September, 6 September, 24 September, 25 September, of 2016, and 31 August, 2017, with the corresponding OA data and model-estimated age displayed on individual altitude-latitude flight track projections for each flight (Figs. S1-S2). A WRF-AAM-derived age example is shown for 24 September 2016 in Fig. S3, at 3 km altitude, in which tracers tagged to CO are released daily from 0-1 day to 7-8 days. Table S1 lists all of the ORACLES-2016 flights and includes comments on their flight pattern, the number of seconds with OA>20 $\mu$g m$^{-3}$ if they were selected for analysis, and otherwise comments on why they were not selected for analysis. Table S2 provides the flight dates, location, time span and altitude of the level legs providing data.

## 2 Sampling Inlet

A high-resolution aerosol mass spectrometer, nephelometers, absorption and soot photometers, and CO/CO$_2$ analyzer were all situated behind a Solid Diffuser Inlet (SDI), with the nephelometers located closest to the inlet and the Aerosol Mass Spectrometer (AMS) and Single Particle Soot Photometer (SP2) located approximately 8m behind the inlet. The SDI brings ambient aerosol into the aircraft and can efficiently transmit aerosol particles smaller than 4$\mu$m in dry diameter (McNaughton et al., 2007). The SDI and ground-sampled submicron scattering data agreed to within 16% during the DC-8 Inlet Characterization Experiment (McNaughton et al., 2007). This establishes the particle loss to the inlet structure, instrument and tubing layout during the ORACLES campaign. Additionally, the sample flow through the inlet was measured and adjusted to ensure the air velocity equaled the flight speed to within 5%. This isokinetic sampling minimizes size-dependent sampling biases (Huebert et al., 1990). Although the inlet was maintained at isokinetic flow, the instruments required a constant flow. An online particle loss calculator (Aerocalc, created by Paul Baron, http://www.tsi.com/uploadedFiles/Product_Information/Literature/Software/ Aerocalc2001.xls) selected tubing material, length, and diameter to minimize particle loss between the SDI and aircraft instruments. The inlet was anodized aluminum, with the flow split into tubes of stainless steel. All lines to the mass spectrometer relied on ½" stainless steel (outer diameter) and ¼" (outer diameter) copper tubing, to reduce the possible presence of extraneous organic compounds. The conductive tubing also minimizes electrophoretic losses. Tubing to the scattering, sizing and counting instruments consisted of graphite-impregnated silicone tubing, with condensation of any released organic compounds upon particles within the air stream unlikely to affect the particle size over the short distance. Due to differences in flow rates and paths, additional losses may affect some instruments more than others. Figures S8-S9 show the plumbing diagram of the aerosol instruments for each year. Calculated losses were negligible, if inherently optimistic and unable to account for all features of the hardware and instruments.

## 3 Wall losses

The ∼8 m distance from the SDI increases the potential for aerosol to deposit upon the tubing wall prior to reaching the AMS. Submicron aerosol is assumed to scatter 5 M m$^{-1}$ at a wavelength of 500 nm per $\mu$g m$^{-3}$ of aerosol mass (Reid et al., 1998; Haywood et al., 2003), termed by them as a mass scattering efficiency. Their result is used to assess if submicron aerosol was lost to wall deposition before reaching the AMS during the ORACLES campaign. The aerosol scattering was measured directly behind the inlet (TSI 3563, 550 nm wavelength) and divided by the total aerosol mass ascertained by the AMS. If the mass becomes depleted by wall losses, the mass scattering efficiency becomes increased. This ratio was evaluated at three different locations/altitudes, resulting in an average scattering by submicron aerosol was 5.92 M m$^{-1}$ per $\mu$g m$^{-3}$. This is close to the expected value of 5, given that the uncertainty in the total aerosol mass concentration is almost 40%. This result constrains the wall losses to within 20% for the entire campaign, though with a wide error margin.

## 4   Aerosol sizing

The SP2 measures rBC diameters between 0.08 and 0.5 micron, with a gradual drop-off in detection sensitivity below 0.08 micron. This effect is evident in the lognormal fits shown in Fig. S4 for level legs from 9/24/2016 and 9/25/2016. The kink at 0.25 micron for the 2016 data samples is an instrument artifact.

The LDMA is heavily modified from a TSI 3071A electrostatic classifier. The flow control, neutralizer and high voltage systems have all been replaced, with only the original DMA column remaining (it is similar to the more recent TSI 3081

DMA). The initial system was modified to scan the voltage. This makes the original system similar to a Scanning Mobility Particle Sizer (SMPS) but the original nomenclature has been maintained here.

The UHSAS optical spectrometer measures particles between 60-1000 nm at a higher one-second time resolution. An infrared laser illuminates particles, with the scattered light collected on two pairs of optical detectors. The particle sizes are then divided into 100 user-specific size bins. The UHSAS undersizes almost 30% of black carbon containing particles during

ORACLES (Howell et al., 2021), because of incorrect refractive indices assumptions ($n_r$=1.588, $n_i$=0). A correction for the undersizing is evaluated in Fig. S6.

Fig. S7 assesses the thermal DMA size distributions for 31 August 2017, 12:14:54, at STP, 150°C and 300°C, done to evaluate the aerosol volatility. As shown, the TDMA size distributions remain consistent regardless of temperature, providing a further indication of low aerosol volatility.

## 5   Aerosol Mass Spectrometer

The AMS sampled the chemical composition of non-refractory aerosols with vacuum aerodynamic diameters between approximately 70 nm to 700 nm at a rate of 1.38 cm$^3$ s$^{-1}$. An aerodynamic lens selects and focuses particles at a constant 600 hPa pressure onto a 650°C heated surface. The non-refractory particles are then evaporated off the heated surface and ionized through electron impaction at 70 eV; the ions are carried forward and analyzed further, with some particles, such as soot,

some organics, dust, and some salts remaining unvaporized (and unanalyzed). A 'V-mode' operation provided a higher time resolution for the same signal-to-noise, with only a modest loss in the mass resolution (see DeCarlo et al., 2006, for more description). The AMS chopper alternately open and closed every two seconds, to allow aerosol into the AMS and to then analyze it, with an additional second separating each duty cycle.

The bulk mass (not size-resolved) chemical species measurements are primarily processed using the SeQUential Igor data

RetRiEval (SQUIRREL, v.1.57l Allan et al., 2003, 2004) data analysis package, with the Peak Integration by Key Analysis (PIKA) program (v.1.16; DeCarlo et al., 2006) resolving the O:C, H:C and OA:OC ratios. Further considerations for the ORACLES AMS-derived aerosol mass concentration data accuracies include the instrument detection threshold, calibrations, and discrimination for organic versus inorganic nitrate. These are considered in that order. Many of the data quality assurance procedures follow those within Shank et al. (2012).

The aircraft-based background values are determined from the noise levels measured at 15,000 ft during a 10-minute time period on the 4 September, 2016 flight. This established detection limits of 0.15 $\mu$g m$^{-3}$ for organics, 0.04 $\mu$g m$^{-3}$ for nitrate, 0.03$\mu$g m$^{-3}$ for sulfate, and 0.01$\mu$g m$^{-3}$ for ammonium. Chloride mass, which nominally contributed $< 1\%$ of the total free-tropospheric aerosol mass, was excluded as its ionization signature varies strongly with composition and exact instrument configuration. Detection limits typically improve during a flight as the background material becomes effused. The AMS was

heated pre-flight during the 2016 campaign to eliminate material built up in between flights. During the 2017 campaign, an initial high-altitude remote sensing leg provided time to drive off extraneous material before beginning the in-situ sampling.

The AMS was calibrated twice during the 2016 campaign (at the beginning and end), and after every 2-3 flights during the 2017 campaign for a total 8 calibrations, using ammonium nitrate particles. An ammonium nitrate solution is sent through an atomizer to produce desiccated submicron aerosol that is then sent to the AMS. A long differential mobility analyzer (LDMA)

(heavily modified from a TSI 3071A electrostatic classifier) selects for 300 nm diameter particles, and a condensation particle counter (TSI 3010) measures the aerosol number concentration. The ammonium nitrate aerosol is diluted by a factor of four in the atomizer to create a calibration curve. The ionization efficiency (IE) of nitrate is thereafter calculated from the aerosol mass and number concentrations. The ionization efficiency estimates the number of ions from a known amount of mass entering the

AMS using the ion signals at *m/z* peak 30 ($NO^+$) and 46 ($NO_2^+$). The nitrate IE values centered on 1.31e-7, with a nominal 10% uncertainty assigned to it following Bahreini et al. (2009), slightly higher than the 1e-7 value within Alfarra et al. (2004). The ionization efficiencies for ammonium, sulfate and organics relative to those for nitrate are thereafter determined within SQUIRREL as: 4 for $NH_4$; 1.1 for measured nitrate relative to the calibration value; 1.2 for $SO_4$; and 1.4 for organics.

A time- and composition-dependent collection efficiency (CE) corrects for the incomplete vaporization of mixed phase particles (Middlebrook et al., 2012), as liquid aerosol is less likely to bounce off the heater and more likely to escape detection than is neutralized aerosol (Huffman et al., 2005; Drewnick et al., 2005). Liquid aerosol is primarily acidic, and the acidity of the free-tropospheric aerosol is assessed by comparing the molar ratio of $NH_4$ to $NO_3+2SO_4$ (Fig. S10). This is a simplification of the $NH_{4,measured}/NH_{4,predicted}$ relationship put forth in Zhang et al. (2007), with the contribution of chloride neglected because it is small. $NH_{4,predicted}$ is the amount of ammonium required to neutralize the inorganic anions observed by the AMS. The applied collection efficiency, $CE = max(0.5, 1 - \frac{NH_4}{2SO_4})$, also neglects the small nitrate contribution, and establishes 0.5 as the lower limit, consistent with most field campaigns (Middlebrook et al., 2012). The ratio of the measured ammonium to the molar sum of nitrate and 2*sulphate is mostly below 1, but rarely below 0.75 (Fig. S10), typically establishing a CE of 0.5. The mildly acidic aerosol suggests mild suppression of inorganic acid formation. Wu et al. (2020) report nitrate aerosol that is fully neutralized based on independent AMS measurements from August-September 2017 further west of the ORACLES sampling, above Ascension Island (8°S, 14.5°W). This indicates further loss of the organic nitrate may be occurring between the ORACLES and Ascension locations. The CE values for the other species are set to 0.5; Middlebrook et al. (2012) do not find any dependence of the CE on the mass fraction of organics.

The overall uncertainty to the reported aerosol mass concentrations is likely dominated by the uncertainty in CE, with additional uncertainty in the organic RIE. Fig. S5a shows the OA:BC mass ratios as a function of model age for OA > 3 $\mu$g m$^{-3}$ (blue) and OA >20 $\mu$g m$^{-3}$ (black), while Fig. S5b shows percentiles of the OA:BC mass ratio composited by aerosol mass bins. Fig. S11-S12 shows the nondimensional BC:$\Delta$CO as a function of *f44* for each flight with sufficient OA > 20 and 10 g m$^{-3}$, and Figs. S13-S14 shows the corresponding OA:BC values.

## 6   Gas measurements

Carbon monoxide was measured with an aircraft modified gas-phase CO/CO$_2$/H$_2$O Analyzer from Los Gatos Research, operated and analyzed by NASA Ames (Jim Podolske). The analyzer uses a patented Integrated Cavity Output Spectroscopy (ICOS) technology to make stable cavity-enhanced absorption measurements of CO, CO$_2$, and H$_2$O in the infrared spectral region. The instrument reports CO mixing ratio (mole fraction) at a 1-Hz rate based on measured absorption, gas temperature, and pressure using Beer's Law (Zellweger et al., 2012). The measurement precision is 0.5 ppbv over 10 seconds.

## 7   Background on the optical measurements

The nephelometer measurements occurred at 40-50% relative humidity. Ambient RH measurements ranged up to 80%, with higher RH data samples excluded by construction. The humidity impact on the nephelometer measurements was examined using two other single-wavelength (550 nm) nephelometers (Radiance Research, M903) measured at two different relative humidities, one at 80% and the other at below 40% RH (Howell et al., 2006). The impact on light scattering, estimated from the ratio of the ambient to dry RH measurements, is estimated to be less than 1.2 for 90% of the time (Shinozuka et al., 2020). The 20% increase in scattering by the ambient RH is an upper bound, as the ambient RH is typically <80%. The nephelometer filter-based measurements are corrected according to Anderson and Ogren (1998).

The PSAP measurements measured at a lower $\sim$ 20% RH, brought about by heating the PSAP optical block to approximately 50°C. The absorption coefficients ($\sigma_a$s) are an average of two PSAP measurements in 2016, with only one PSAP functioning in 2017. Corrections to the absorption coefficients are based on the wavelength-averaged (as opposed to wavelength-length-specific) values from Virkkula (2010). The use of the average wavelength-corrected values reduces a potential high bias from multiple scattering at the shortest wavelength (Pistone et al., 2019), and reduces spurious effects from filter changes (Zuidema et al., 2018). Compared to Pistone et al. (2019), a stricter aerosol threshold is applied (OA>20 $\mu$g m$^{-3}$ rather than scattering at 530nm > 10 Mm$^{-1}$) and no arithmetic weighting by extinction is done. SSA values at 530 nm are at standard temperature and

pressure. Aerosol absorption can also increase because of humidification (see discussion in Pistone et al. (2019)), introducing a compensating effect on the SSA, but this is likely smaller.

Two nephelometers (TSI 3565), located near the aerosol inlet, were also used to assess the contribution of submicron aerosol to the total aerosol scattering. One nephelometer measured only the submicron scattering, while the second nephelometer measured both total and submicron scattering. The measured total to submicron scattering ratio was 1.02, confirming that almost all of the free-tropospheric scattering is by submicron aerosol.

*Author contributions.* The present work was conceived by PZ, AD, SH and PS. SF contributed to the HiGEAR data analysis, AS provided
the BC datasets and PS the WRF-AAM model age estimates. Portions of this work first appeared in the M.S. thesis of A.D at U. of Hawaii. PZ led the writing and AD provided most of the figures, with all authors contributing to the final writing.

*Competing interests.* Paquita Zuidema is a guest editor for the ACP Special Issue: "ACP special issue: New observations and related modelling studies of the aerosol–cloud–climate system in the Southeast Atlantic and southern Africa regions" The other authors declare no competing interests.

*Acknowledgements.* ORACLES is a NASA Earth Venture Suborbital-2 investigation, funded by the US National Aeronautics and Space Administration (NASA)'s Earth Science Division and managed through the Earth System Science Pathfinder Program Office (grant no. NNH13ZDA001N-EVS2). This work was further supported by the US Department of Energy (DOE: grant DE-SC0018272 to P.Z. and P.S. and DE-SC0021250 to P.Z.).

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

**Table S1.** ORACLES flights occurring between August 31 and September 31. Examined flights in bold.

| Date (M/DD/YYYY) | Flight label | Flight Description | sec. with OA>20 $\mu$g m$^{-3}$ and BC, CO, SSA |
|---|---|---|---|
| 8/27/2016 | PRF00Y16 | transit | probes off |
| 8/30/2016 | PRF01Y16 | routine | aborted |
| **8/31/2016** | PRF02Y16 | routine | **3,447** |
| 9/02/2016 | PRF03Y16 | target | no OA>20, 6,341>10 $\mu$g m$^{-3}$ |
| **9/04/2016** | PRF04Y16 | routine | **1,760** |
| **9/06/2016** | PRF05Y16 | target | **3,765** |
| 9/08/2016 | PRF06Y16 | routine | aerosol age>10 days (too old) |
| 9/10/2016 | PRF07Y16 | routine | aerosol age>10 days (too old) |
| 9/12/2016 | PRF08Y16 | routine | 38 |
| 9/14/2016 | PRF09Y16 | target | 161 |
| 9/18/2016 | PRF10Y16 | target | no BC data |
| 9/20/2016 | PRF11Y16 | target | no OA>20, 2,840>10 $\mu$g m$^{-3}$ |
| **9/24/2016** | PRF12Y16 | target | **4,072** |
| **9/25/2016** | PRF13Y16 | routine | **2,732** |
| 9/27/2016 | PRF14Y16 | transit | probes off |
| **8/31/2017** | PRF12Y17 | target | **11,743** |

**Table S2.** Level legs

| Flight | Latitude °N | Longitude °E | Time (UTC) | Altitude (m) |
|---|---|---|---|---|
| 08312016 (PRF02Y16) | -13.8 :-13.13 | 3.7:3.9 | 11:14-11:27 | 3830 |
| 09062016 (PRF05Y16) | -12.9:-12.2 | 9.2:9.5 | 11:40-11:50 | 2670 |
| 09062016 (PRF05Y16) | -15.9:-15.17 | 10.3:10.5 | 12:18-12:28 | 2250 |
| 09242016 (PRF12Y16) | -12.1:-11.0 | 11.0 | 12:08-12:21 | 4830 |
| 09252016 (PRF13Y16) | -9.8:-8.9 | -0.344:-1.0 | 12:16-12:32 | 4500 |
| 08312017 (PRF12Y17) | -8.6:-7.5 | -1.27:-0.88 | 11:43-11:57 | 3100 |
| 08312017 (PRF12Y17) | -7.5:-6.7 | -1.58:-1.3 | 11:31-11:42 | 3035 |
| 08312017 (PRF12Y17) | -6.4:-5.6 | -1.99:-1.65 | 11:15-11:30 | 2935 |
| 08312017 (PRF12Y17) | -5.29:-4.12 | -1.64:- 2.01 | 10:52-11:11 | 2870 |
| 08312017 (PRF12Y17) | -8:-5.12 | -1.09:-2.15 | 12:12-12:50 | 2970 |
| 08312017 (PRF12Y17) | -2.5:-0.5 | -0.44:0.8 | 14:10-14:34 | 2880 |
| 08312017 (PRF12Y17) | -1.9:-1.23 | -0.105:-0.404 | 1:55-14:05 | 2720 |
| 08312017 (PRF12Y17) | -2.4:-1.35 | -0.70:-0.15 | 13:32-13:48 | 2570 |
| 08312017 (PRF12Y17) | -3.8:-1.27 | -0.12:-1.49 | 10:12-10:49 | 2790 |

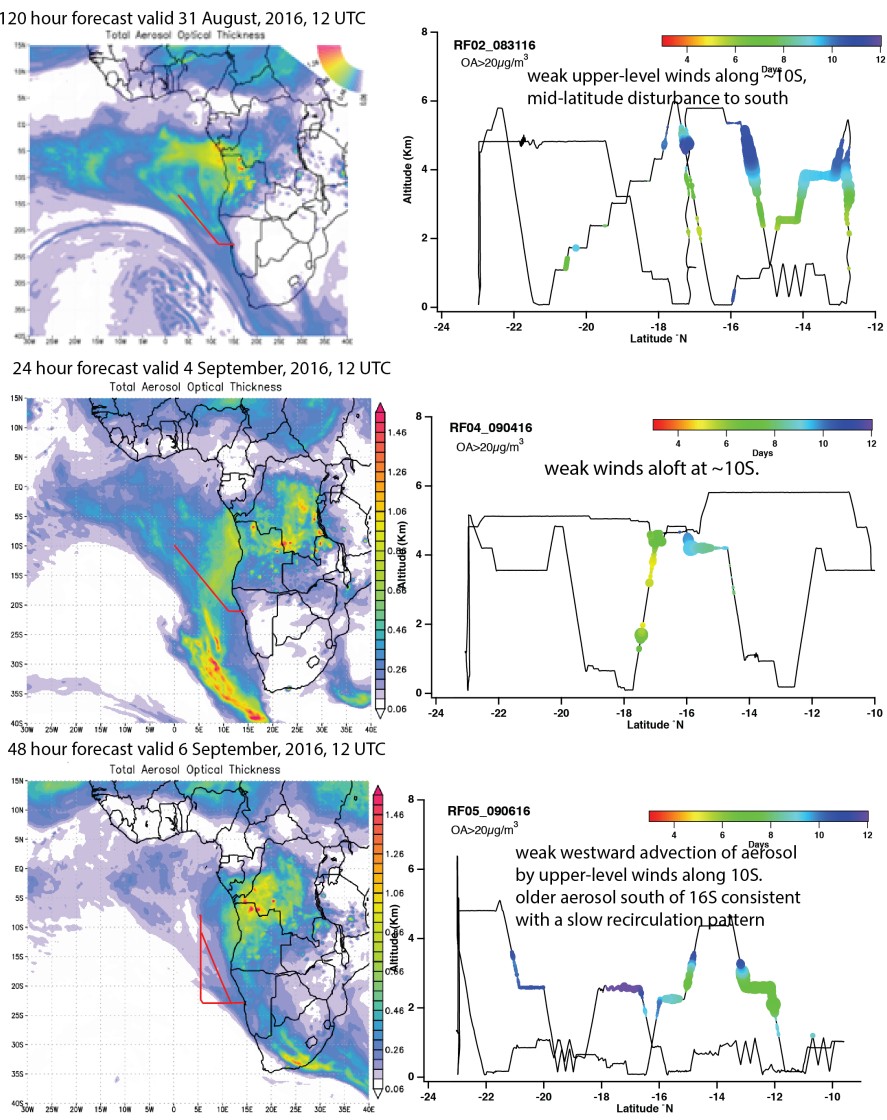

**Figure S1.** left) Global Modeling and Assimilation Office aerosol optical thickness forecasts for 31 August, 2016; 4 September, 2016; 6 September, 2016. Right) Altitude versus latitude cross-sections of the flights overlain with the colorized aerosol age, with the size of the marker providing a qualitative marker of aerosol mass.

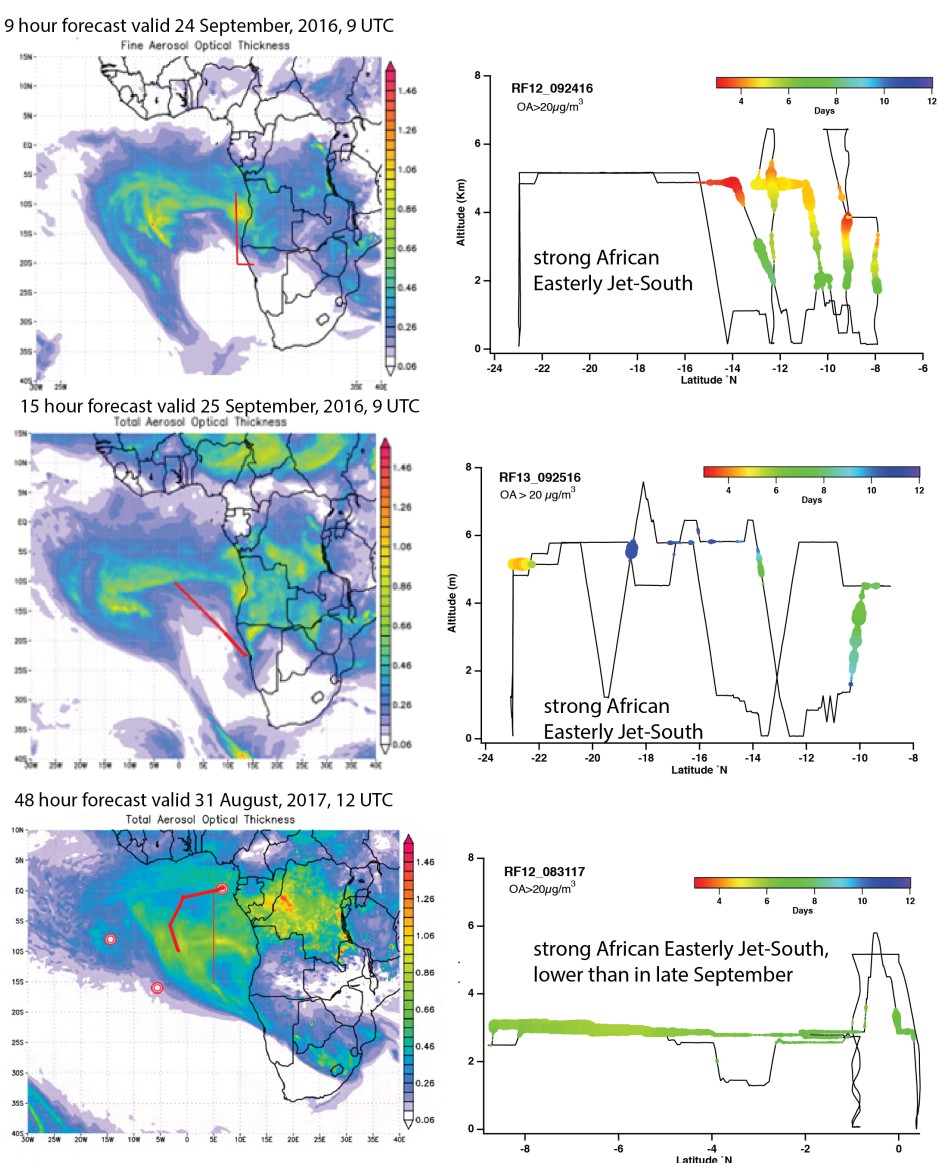

**Figure S2.** left) same as Fig. S1 but for 24 September, 2016; 25 September, 2016, and 31 August, 2017.

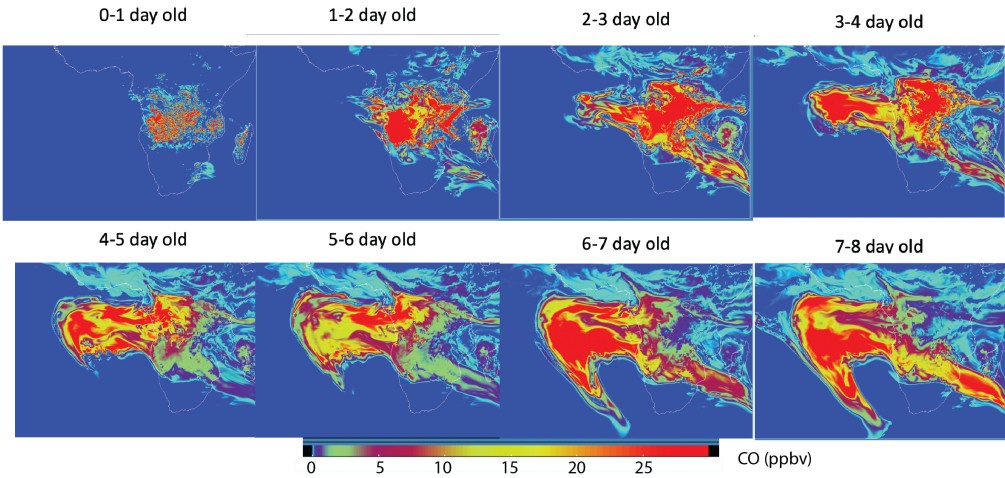

**Figure S3.** 24 September 2016 WRF-AAM CO-tracer from 0-1 up to 7-8 days since emission, at ∼ 3 km altitude. CO units in ppbv, with zero background CO.

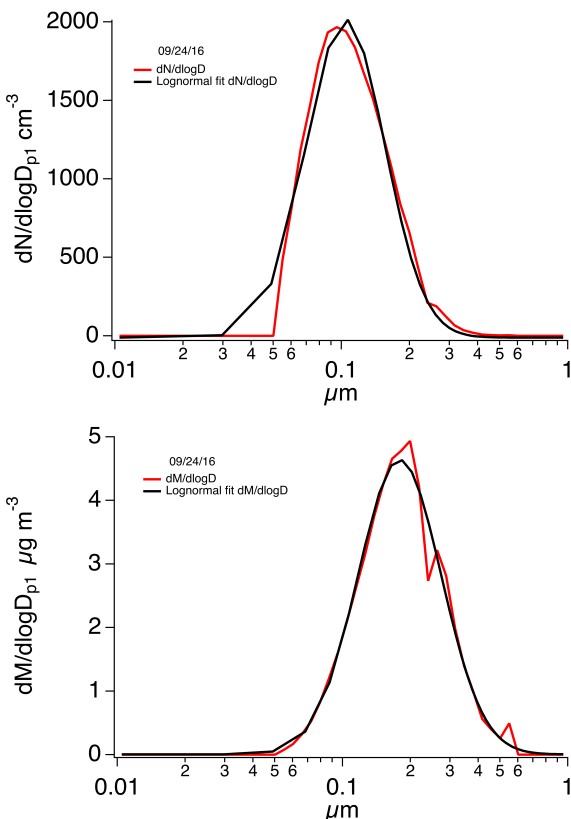

**Figure S4.** Lognormal fits (black) to the measured SP2 mass distributions (red) by size for top) 9/24/2016 and bottom) 9/25/2016 level legs shown in Table S2.

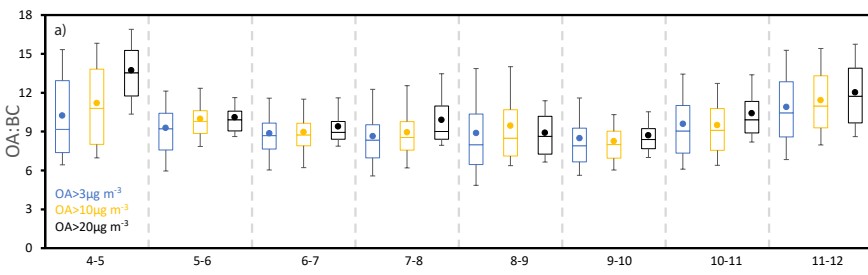

**Figure S5.** OA:BC mass ratio as a function of model age (in days) for OA mass concentrations>3 $\mu$g m$^{-3}$ (blue), > 10 $\mu$g m$^{-3}$ (yellow) and >20 $\mu$g m$^{-3}$ (black).

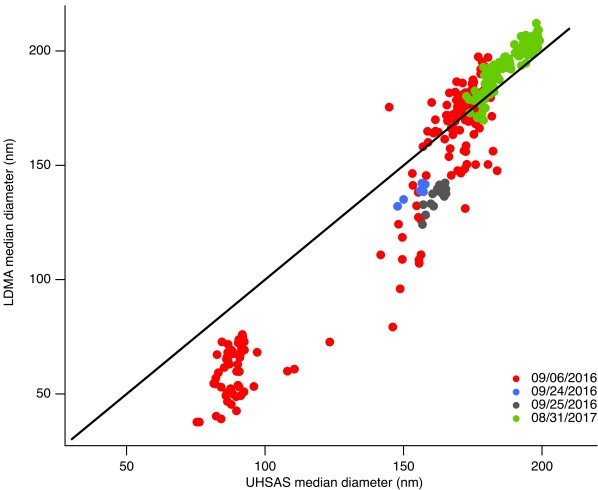

**Figure S6.** LDMA versus UHSAS median diameters for samples within the level-leg plumes (Table S2) with OA $>20$ $\mu$g m$^{-3}$, at one-minute time resolution.

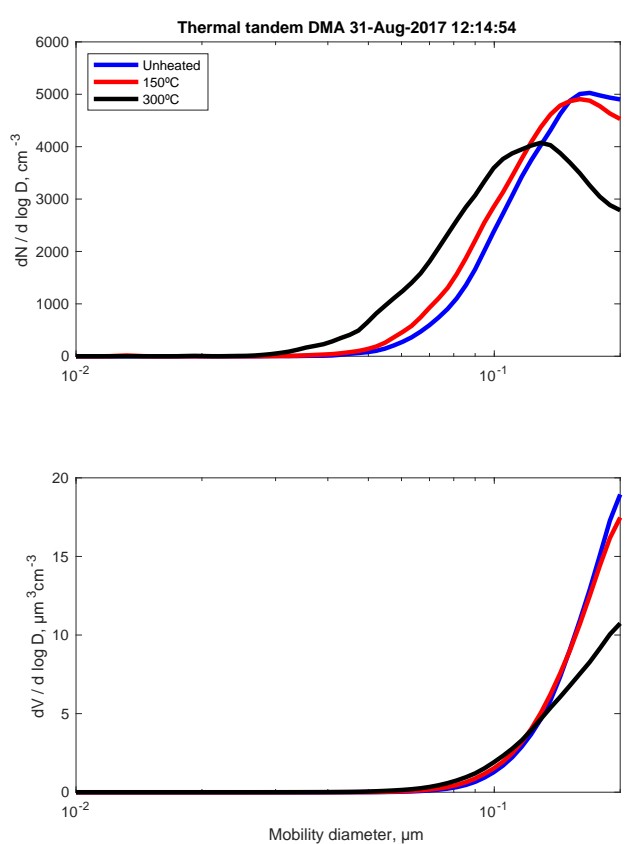

**Figure S7.** TDMA number (top) and volume (bottom) size distributions as a function of standard temperature and pressure (blue), heated to 150°C (red) and 300 °C (black), for 31 August 2017, 12:14:54.

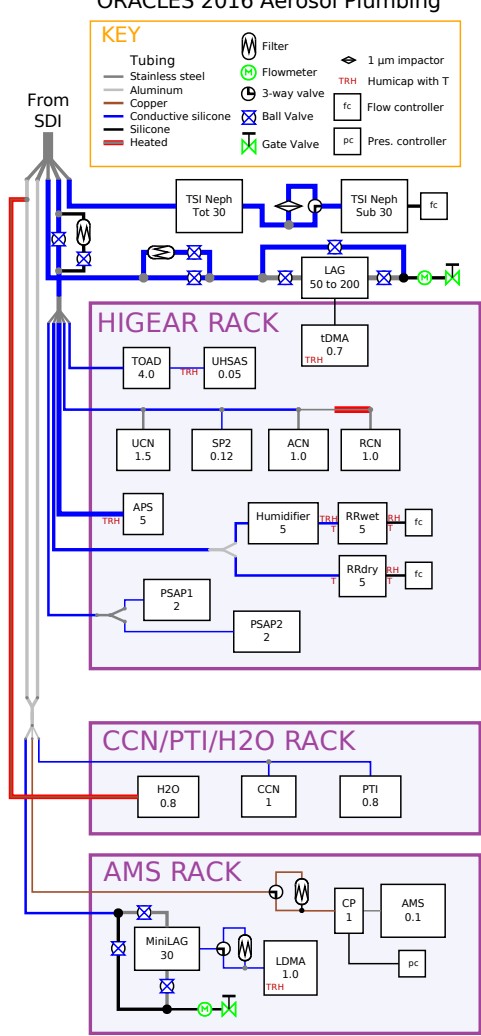

**Figure S8.** Layout of aerosol instrumentation relative to the inlet for the 2016 campaign. The numbers below the instrument acronyms represent flow rates in lpm. Note the lag and mini-lag include a small leak included to equalize the pressure between the two. The line widths are proportional to the nominal diameter of the tubing (outer for metal, inner for silicone). Exceptions are the AMS, SP2, and UHSAS, which have very low flow rates and such tiny inlet tubes that they wouldn't be visible. Acronyms, in alphabetical order: ACN=Ambient Condensation Nuclei (unheated TSI 3010); APS=aerodynamic particle sizer (TSI; LAG=Lagged Aerosol Chamber; CP=Constant Pressure inlet to the AMS. RCN=Refractory Condensation Nucleus Counter (TSI 3010, operated at $400^\circ$C); $RR_{wet}$=humidified Radiance Research nephelometer; $RR_{dry}$=Radiance Research nephelometer at 550 nm wavelength and low relative humidity; TOAD=thermo-optical aerosol discriminator; UCN=Ultrafine Condensation Nucleus Counter (TSI 3025, diameters $> 2.5$ $\mu$m). Other acronyms are described within the main text. Not all of these instruments were used for this analysis, and a particle cavity aerosol spectrometer probe (PCASP) operated by U of North Dakota is not shown here.

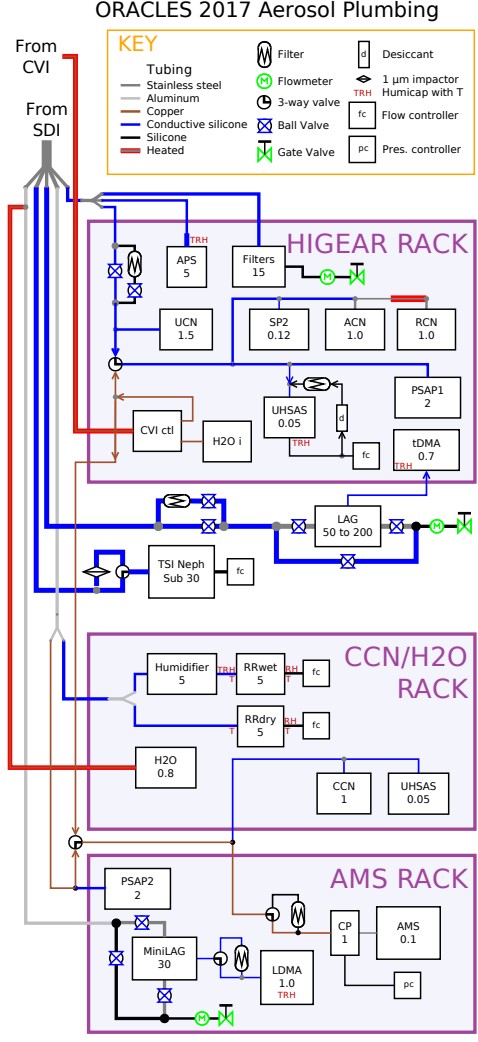

**Figure S9.** Layout of aerosol instrumentation relative to the inlet for the August 31, 2017 flight. Most flow is down and to the right, the addition of a counter-flow virtual impactor inlet (CVI) modified some flow direction to be up and to the left; flow direction arrows are included in critical spots to aid understanding. Other comments on the diagram Fig. S8 for 2016 also apply here.

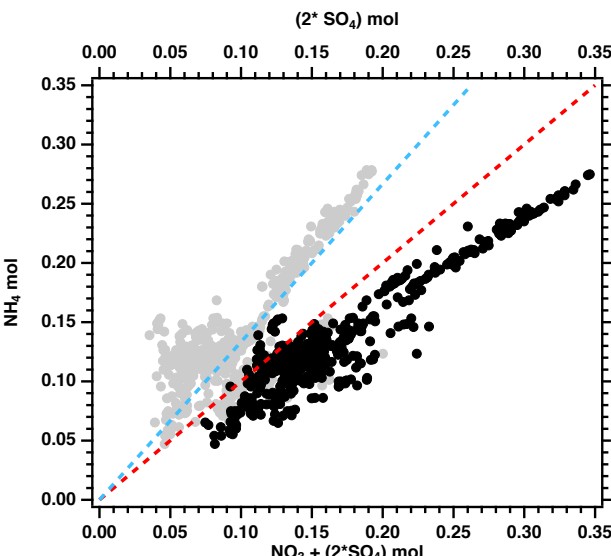

**Figure S10.** Measured ammonium in moles as a function of the molar sum of nitrate and 2*sulphate for one-minute measurements from all 6 flights (one-minute averages), constrained to the free troposphere. Dashed-red and dashed-blue lines indicate the 1:1 and 1:0.75 ratios respectively.

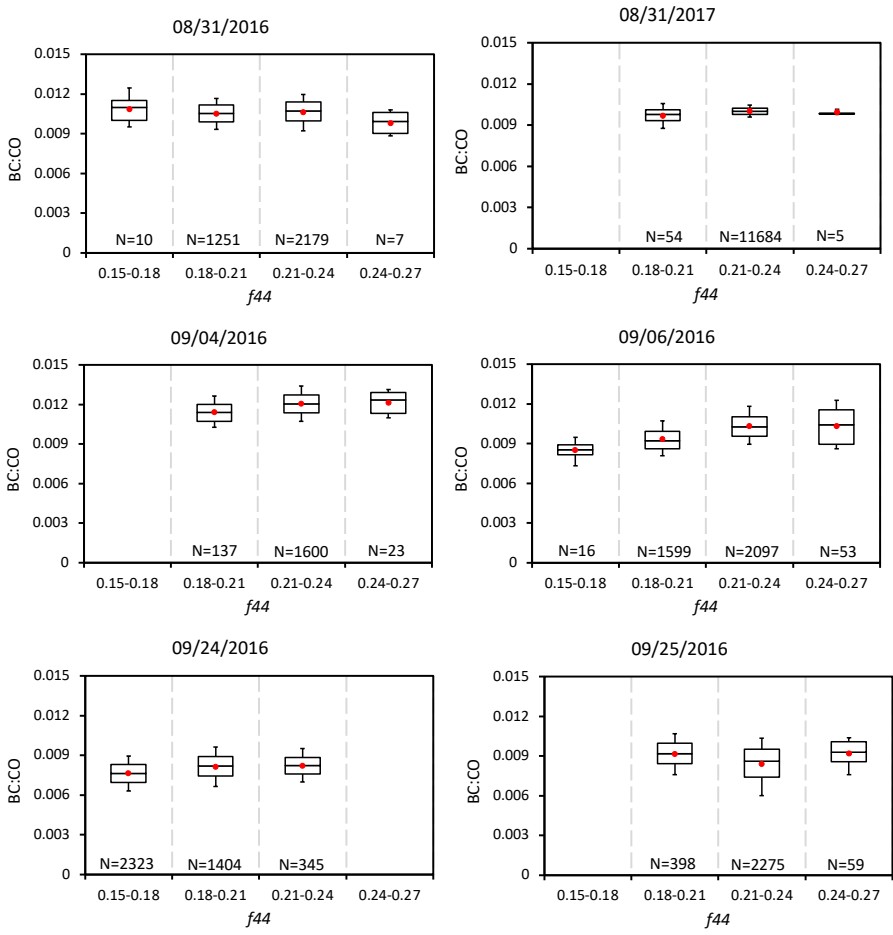

**Figure S11.** BC:ΔCO ratios (dimensionless) as a function of *f44* for the six flights. Whiskers represent the 10th and 90th percentiles, boxes illustrate the 75th and 25th percentiles with a line indicating the median and a red filled circle the mean. OA>20 $\mu$g m$^{-3}$ only. The number of 1-second samples contributing to each *f44* bin of each flight is also indicated.

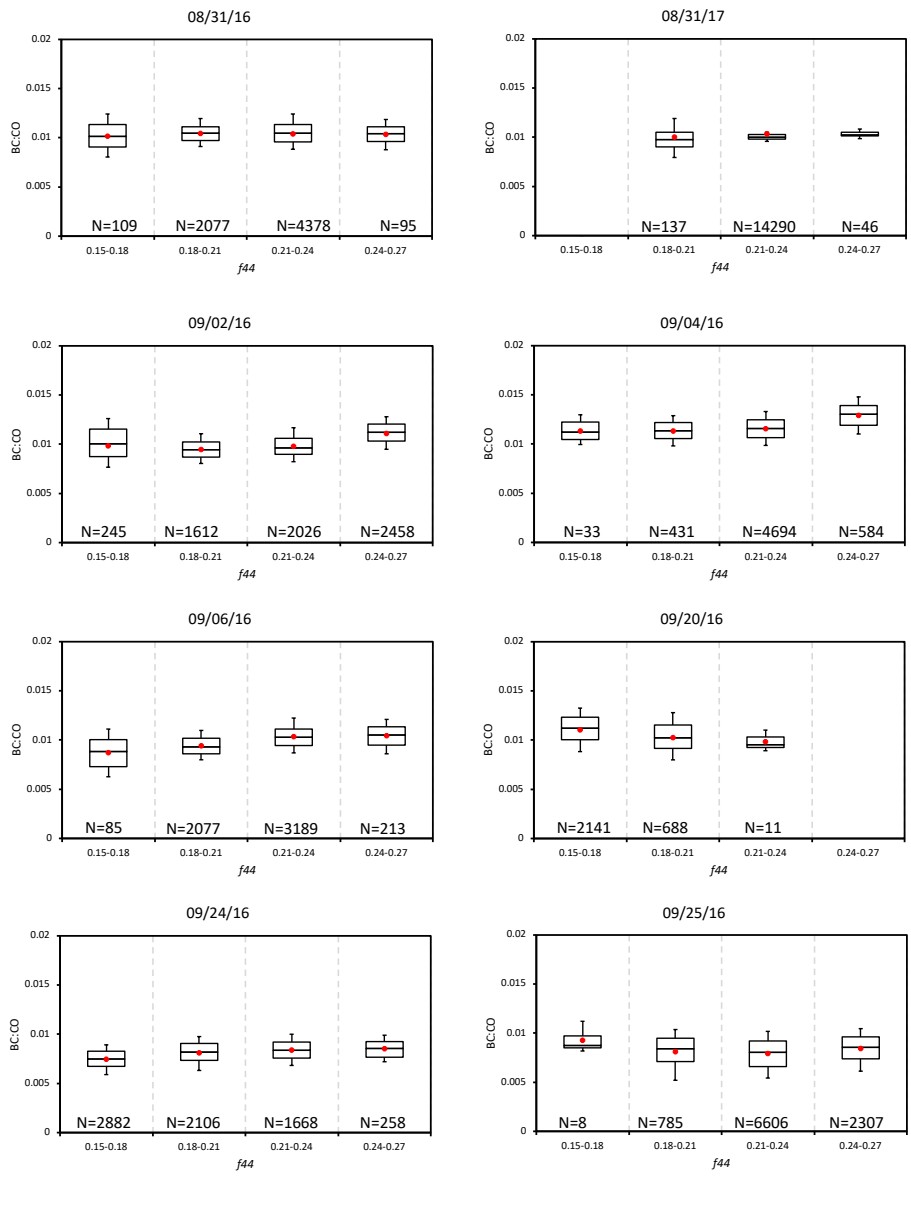

$OA > 10\mu g\ m^{-3}$

**Figure S12.** BC:$\Delta$CO ratios (dimensionless) as a function of *f44* for the six flights. Whiskers represent the 10th and 90th percentiles, boxes illustrate the 75th and 25th percentiles with a line indicating the median and a red filled circle the mean. OA>10 $\mu$g m$^{-3}$ only. The number of 1-second samples contributing to each *f44* bin of each flight is also indicated.

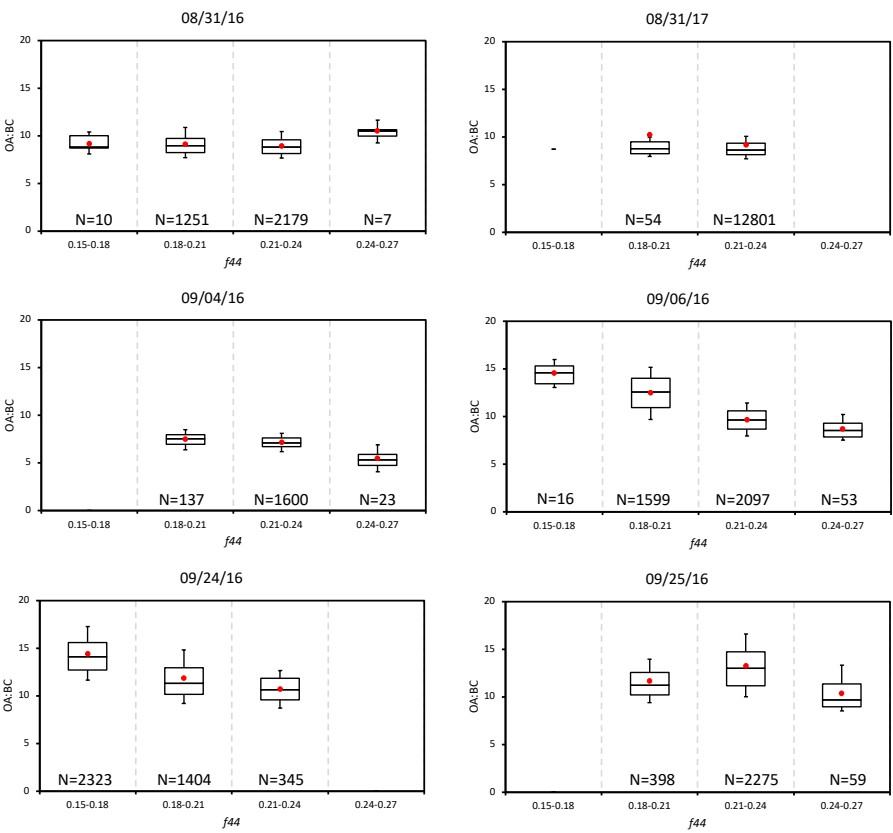

**Figure S13.** OA:BC mass ratios as a function of *f44* for the six flights. Whiskers represent the 10th and 90th percentiles, boxes illustrate the 75th and 25th percentiles with a line indicating the median and a red filled circle the mean. OA>20 $\mu$g m$^{-3}$ only.

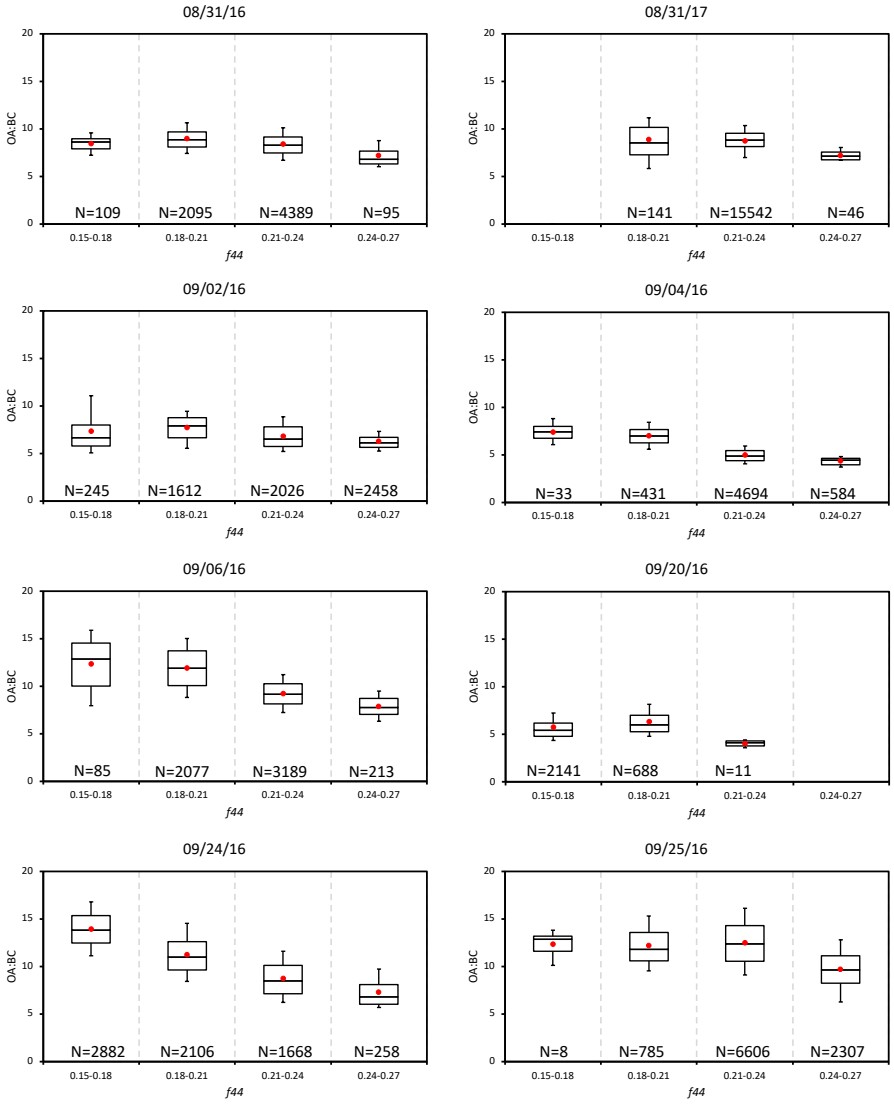

$OA>10 \ \mu g \ m^{-3}$

**Figure S14.** OA:BC mass ratios as a function of *f44* for the six flights. Whiskers represent the 10th and 90th percentiles, boxes illustrate the 75th and 25th percentiles with a line indicating the median and a red filled circle the mean. OA>10 $\mu$g m$^{-3}$ only.