# Peer review of "An attribution of the low single-scattering albedo of biomass-burning aerosol over the southeast Atlantic"

_Atmospheric Chemistry and Physics, 2022_

## Author Comment (AC1)

**Response to Reviewers** of ACP-2055-510 "An attribution of the low single-scattering albedo of biomass-burning aerosol over the southeast Atlantic" by A. Dobracki et al.

We appreciate the time the reviewers have invested in assessing this manuscript and have done our best to address the points raised. We go through these individually below, with the reviewer comments verbatim in blue and our responses in red. Manuscript text, where included, is done in black.

**Reviewer 1:**

This paper provides new insights on this hot topic. However, I do have some comments related to the readability of the paper and the interpretation of the results that I feel are critical for the authors to address prior to acceptance.

- The paper is very difficult to read because it contains many mistakes and writing problems. All co-authors must carefully read the manuscript and correct any errors. It is not possible to list here all the errors, but it includes:
  - missing figure number (e.g. L421) and figure number in the wrong sequence (e.g. Fig. S7-S8 before Fig. S6, Fig. S4 before Fig. S3,…).
  -

We were unclear if the figures in the supplement should be numbered sequentially as they appear in the supplement, or, if their numbering should follow how they are called out in the main text. We consulted with Copernicus on this, and were told that the figures in the supplement can be arranged as we prefer, as long as they are labeled using the 'S' (for Supplement). We have already followed this. In light of this comment, however, an effort has been made to rewrite the supplement to have the figures follow the order in which they are called out in the main text.

We will do a careful read-through, enlisting all the co-authors, before resubmitting the manuscript. The second reviewer also commented on the poor readability, and we do want to make sure to fix that as it will encourage a wider readership….

  - errors (eg. Fig. 9 is colored by flights instead of AAE and MAC values), missing elements (eg. the name of y-axis in Fig. 4b, the unit of y-axis in Fig. 7a, 7d, 12a and 12b) and readability issues (eg. Fig. 12c and 12d) in some figures.
  -

Fig. 4b: name fixed in caption.
Fig. 7a,d: units added to the y-axis labels.
Fig. 9: The idea with coloring the locations by the flight is that it allows the reader to cross-reference the values shown in Fig. 8 with the locations, affirming that transport away from the continent is associated with an increase in the black carbon containing particle number, decrease in SSA, etc. We wish to hold on to this colorization scheme. We did increase the font size on the AAE/MAC values to improve their legibility.
Figs. 12c and d: panel sizes increased.

We have gone through the figures one by one to address other readability issues, primarily with font sizes that are too small.

- inconsistency in the abbreviation/acronym usage (eg. the use of "black carbon"/" BC", "brown carbon"/"BrC ", "organic carbon"/"OA", "modified combustion efficiency"/"MCE"… at different places in the text)

We have read through the manuscript again to make sure the acronym usage remains consistent and have removed multiple definitions of acronyms.

- Some parts of the Introduction includes some description of the flights and a brief presentation of the results that are very difficult to follow without reading the entire document. The choice of the flights analyzed in the paper should be moved in Section 2 after the presentation of the ORACLES and CLARIFY field campaigns. The authors could introduce the results sections in a more classical way to make reading easier: "Section 4 presents the chemical composition and age distribution within the six flights", "Section 5 investigates the link between BBA optical properties and chemical composition",… The last Section 8 is also difficult to read because it follows very dense discussion sections.  A brief summary of the main results would help the reader.

Reviewer 2 also mentioned this material didn't belong in the introduction. We have moved the language about the flights to section 2, introduce the sections as suggested by the reviewer, and moved/consolidated some of the brief discussion of the results to the section 8 as part of results summary within that later section.

- The authors interpret the changes in BBA composition and optical properties to fragmentation of oxidized aerosol thought photochemistry. This is based on the observed decrease in OA mass concentration and increase in oxidation with plume age. I am wondering if other processes can't explain these results. Do you have elements to reject the following assumptions?

  - Aqueous phase reaction and cloud processing could contribute to the oxidation of OA and decrease in OA mass concentration.

All measurements were made in the free troposphere, at RH<80%, to remain outside of any mid-level clouds. The convection over land is almost exclusively dry (see Ryoo et al., 2021; 2022), but the top of the boundary layer over land can reach humidities capable of supporting clouds (e.g., Adeyemi et al., 2020). We cannot conclusively ignore aqueous phase reaction nor cloud processing. We have included language on line 414 (original document) to that effect: "Aqueous phase reactions and mid-level cloud processing could potentially also contribute to the oxidation and free-tropospheric OA mass concentration decrease. This is partially controlled for by only selecting data samples with RH$<$80\%. Mid-level clouds, produced by dry convection saturating the top of the land boundary layer, can occur \citep{Adebiyi20}, but are not a dominant presence on this day or other ORACLES flight days. This suggests to us that the reduction of free-tropospheric OA through aqueous phase reactions is of secondary importance (becoming even more with distance from the continent)."

  - The condensation of less oxidized material onto preexisting highly aged organics favored by lower temperature during transport would favorize the evaporation of OA into the gas phase.

True. This process could also be contributing. We now mention this process at the end of section 7 (original line 474), where we discuss the thermodynamical repartitioning. The additional sentence is "At higher altitudes, less-oxidized material may also continue to condense upon the pre-existing organics, ultimately favoring the evaporation of OA into the gas phase."

Specific comments:

L9: I can't see the link between the increasing fraction of BC-containing with chemical age, and the processes of vapor condensation and coagulation. Please clarify it. We just used chemical age as a proxy for time here and can see how that is confusing. We have rewritten the sentence as "The particle size and fraction of BC-containing particles increased over time, consistent with ongoing gas condensation and the coagulation of smaller non-BC particles upon the BC-containing particles. " Maybe I am missing something else the reviewer is puzzling over - the BC particles are smaller than the non-BC and it is primarily the smaller particles that become less numerous. An additional phrase points this out.

L12-13: It would be clearer for the reader to add that BBA sampled during CLARIFY have travelled longer distance than those sampled during ORACLES. We have rewritten this as: The CLARIFY (CLoud-Aerosol-Radiation Interaction and Forcing: Year-2017) aircraft campaign sampled aerosols that had traveled further to reach the more remote Ascension Island, reporting higher BC number fractions, lower OA:BC mass ratios, lower SSA yet larger mass absorption coefficients compared to this study's.

L15-17: The reason to focus on inorganic ammonium nitrate is not clear when reading the abstract. There would have been insufficient purely scattering nitrate particles to explain the vertical variation in the SSA? We decided to leave the language on ammonium nitrate out of the abstract after making the other changes. Another criticism of using nitrate partitioning to explain the SSA vertical structure is that there actually was little nitrate overall in the free troposphere above Ascension (<10%).

L17 : Please remove « 2017 » We have removed this sentence.

L103: I am wondering if $CO_0$ should not decrease with the altitude due to vertical dilution. Why did not you use CO measured outside BBA plumes to obtain the background values as a function of the altitude? Pristine air at the same altitude was rarely sampled, as the flight sampling objective was to sample the aerosol layers on 3 of the flights, and in the 3 'routine track' flights….we also note that the large-scale subsidence typifying this region introduces stability gradients that discourage the vertical dilution of $CO_2$

L134: please replace "later" by the corresponding section. We have rewritten this paragraph so that this phrase no longer appears.

L168: The manufacturers of CPC and SP2 are missing. The SP2 manufacturer (DMT) was previously identified on the original line 106-107. The CPC manufacturer was identified later on line 187, we have moved this information to line 168 where it should be.

L177: Which refractive index did use for UHSAS corrections? n_real=1.588, n_i=0. Now included in the supplemental text as the second reviewer also asked.

L181: Please add P=1013 mbar done.

L197: Do you mean that you used the scattering Angtrom exponent to convert scattering coefficients to different wavelengths? Yes. We've rewritten this though to just say the scattering measurements are interpolated to the PSAP wavelengths.

L209: Why did you choice the limit of OA > $20 \mu$g m-3? Could not it biases your results? (For instance missing analysis of case studies with lower OA to BC ratio)

We go through a lengthly explanation of this in lines 128-143 and include figures using a lower 10 microgram/m^3 threshold within the supplement. The choice is inherently subjective but does serve a purpose. This section now reads as:

"A threshold of 20 $\mu$g m$^{-3}$ is applied to select for the heart of the smoke plumes. This is one approach to minimizing dilution effects, by which OA evaporates through mixing with cleaner environmental air \citep[e.g.,][]{Hodshire21}. This threshold was selected based on when a stabilization of the OA:BC mass ratio occurs as a function of the OA mass concentrations (Fig. S5). The OA:BC mass ratio is significantly less for air with OA$>$3 $\mu$g m$^{-3}$ than for air with OA$>$20 $\mu$g m$^{-3}$, particularly for younger aerosol (Fig. S5a). The choice of threshold is inherently arbitrary, and some analysis is repeated using an OA$>$10 $\mu$g m$^{-3}$ threshold to make sure our findings are not sensitive to the choice of OA mass threshold. An additionally-applied approach to removing dilution effects is to normalize with respect to BC or $\Delta CO$. We stress, however, that the aerosol plumes are typically much larger and homogeneous than those sampled from fires in the western northern hemisphere, which are often named, individual fires sampled close to the source. Dilution of the aerosol plume is much less of a concern during smoke-filled conditions over the southeast Atlantic. The OA mass concentrations are often highly stable during the individual level legs (Table S2), with 20 minutes of data reducing the absolute (relative) uncertainty in the OA mass concentration to at most 1.6 $\mu$g m$^{-3}$ (8\%). Fig. S5a also indicates OA:BC mass ratios can increase again 10 days after emission, but we exclude such old aerosol as the model skill in the smoke age is likely to become less over time. Further justification for application of a threshold is that the uncertainty in the OA mass is smaller at higher signal-to-noise ratios, and, that model-observational disparities in the smoke plume locations have less impact on further aging-related analyses if based on the plume centers. "

The Supplement contains two analyses that are redone using an OA mass threshold of 10 micrograms/m^3 to ensure that none of our findings are dependent on the particular threshold choice.

L232 : Please replace g m-3 by $\mu$g m-3. Done

L235-236 : Does it mean that BBA were not dominated by OA during CLARIFY ? CLARIFY BBA was also dominated by OA. This sentence has been rewritten as "This [OA mass] is much more than measured in the free troposphere above Ascension during CLARIFY \citep{Wu20}, although the OA mass fraction during CLARIFY still remained > 50%."

L250-252 : The authors could mention that BC was not measured by the same method  (SP2, thermal-optical transmission) in the literature summary in Table 1, which may explain some differences. We mention this as a table footnote ("Methods for deriving the BC mass concentration may vary between the studies") as regardless the ORACLES September BC/ Delta(CO) values remain among the highest surveyed.

L 271: Table 1 instead of Table 2 done

L277: Could you please remind the reader that the peak at m/z 60 is often associated with levoglucosan from biomass burning? Done

L285: I don't understand why O:C values in BBA are expected to be comparable with urban measurements at Mexico ? The MILAGRO campaign also characterized biomass burning, for which they could do a confident source attribution. That said, the sentence does seem out of context upon rereading it, and we have removed it.

L311: Please precise that measurements were performed in the remote South West Africa in Denjean et al (2020) and provide the range of BC-containing particle faction obtained in Taylor et al. (2020). Done

L335-336: I don't understand what you mean by "because our data lack highly-scattering aerosol". Replaced with: "primarily because our dataset has a smaller SSA range, with no SSAs > 0.9. "

L343: Please provide a reference for the primary emission of BrC. This sentence was overstated (e.g., Forrister et al., 2015) and removed.

L353: AAE is in the range 1.1-1.3. Does it mean that BrC is a significant contribution to BBA absorption ? No. The Forrister 2015 is a good reference for this. We now mention this in the text.

L383 : Please precise that precursor gases may be more avalaible for nucleation. Understood. done.

L385: Does a constant dBC/dCO mean that BBA had the same source and that there was no wet deposition ? We are interpreting it that way, and now mention that within the text.

**Reviewer 2:**

The subject is of high interest and the complexity of this topic requires a high level of understanding in the interaction of physical and chemical processes. Some interesting aspects are raised in the present manuscript. However, there are some deficiencies in content and presentation. These should be addressed before the manuscript can be accepted.

General comments:

The manuscript is difficult to read, as one must read the appendix in parallel with the manuscript to understand it. The description of the aerosol devices and experimental setup are not sufficiently described and difficult to read. In some cases the common abbreviations are not used. Furthermore, an error analysis due to experimental uncertainties and instrumental artefacts is missing. A summary of measurement techniques and derived parameters in tables would increase the readability.

We have now included a table listing the measurements and derived parameters, thanks for the suggestion. We do include an error analysis e.g. on the AMS-derived quantities, and would need more information to better understand what is missing. It's true that some of it is relegated to the Supplement, as there is simply too much material on the measurements to put it in the main text. Upon rereading we do see how the manuscript was difficult to read, and have rewritten with more sensitivity to how the manuscript might be perceived by someone encountering it for the first time.

Analyses and discussion on optical properties is driven by correlations of observed mass and number fractions. A discussion using mixing models and light scattering theory is not mentioned. Even though mixing state data are not available from SP-2 and even though scattered light theory was not the focus of this manuscript, the basic insights derived from light scattering theory, e.g. light absorption enhancement factor, should be considered in the discussion.

It's true that the current discussion does not say anything about the optical modeling. We have included a discussion now in a new 2nd paragraph, reproduced below:

The total aerosol concentration exceeds the total SP2-derived BC number by a factor of 2.5 to 7, we can infer that at least some of the non-BC aerosol remains externally mixed with the BC. The BC itself, because of its transport within multiple days within broad, dense smoke plumes, is most likely internally mixed with other aerosols, confirmed by electron microscopy measurements within \cite{Dang22}. Because the BBA is at least 4 days old, and as already shown within \cite{Taylor20} and \cite{Denjean20}, the BC can be treated as compacted. \cite{Taylor20} find a better fit to the CLARIFY MAC measurements using the semi-empirical models of \cite{Liu17} and \cite{Chakrabarty18} than to a core-shell Mie model, but \cite{Lee22} conclude a core-shell mode can be successfully applied once particle-by-particle differences are accounted for. The 2016 data from ORACLES lacks SP2 mixing state date with which to better evaluate optical fits, but an independent assessment could be pursued using the SP2 coating-resolved ORACLES data from 2017 and 2018 \citep{Sedlacek22}.

Specific comments:

Line 20: BBA yet not defined. We have substituted the phrase 'smoke emissions' here

Line 110: Is the limited size range sufficient for the analyses. How large is the fraction of particles not detected? Is this fraction constant or does it change in the different cases?

The SP2 captures mass-equivalent diameters between approximately 80-500 nm.  This size range is expected to successfully capture 99\% of the black carbon mass \citep{Taylor20}. We do not see a reason why this would change between the flights.

Line 167: Were the mobility spectra corrected for multiple-charges? The inversions include a size-dependent charging efficiency and accounts for multiply-charged particles and also considers size-dependent losses. The inversions are taken from Zhou, 2001. This information is now included in the main text.

Reference: Zhou, J.: Hygroscopic properties of atmospheric aerosol particles in various environments, Ph.D. thesis, 166 pp., ISBN 91-7874-120-3, LUTFD2/(TFKF-1025)/1-166/(2001), Lund University, Lund,Sweden, 2001.

Line 176: What refractive index was used for correcting the UHSAS size spectra? $n_r=1.588$, $n_i=0$. Now included in the supplemental text as reviewer 1 also asked.

Line 182 to 184: Why are the problems with PCASP mentioned? If necessary, this would fit better into a chapter on corrections and quality control.

We mention because of the comparison to CLARIFY values later on, which used PCASP total number concentrations where we used those from the LDMA. The reviewer is right though, that the detail is out of place here. We have removed the language from the manuscript but do insert a footnote to the Table 2 about the slight difference.

Line 203: 'Ångström' throughout the manuscript

\AA ngstr\"{o}m in Latex. done.

Line 204: Should be called rBC when measured with SP2

It seems common to call SP2-derived refractory black carbon BC (e.g., Taylor et al., 2020, Denjean et al. 2020) and we follow their lead on this. We do include language in the main text now under Section 2.4 Aerosol Composition to mention that 'throughout we use BC to refer to the SP2-derived refractory black carbon' as we do understand they are not entirely the same. We also cite the Petzold paper.

Lines 310 ff: The authors state, "The mass absorption coefficients (MAC660nm) and SSA values depend to first order on an estimate of the fraction of particles containing black carbon." This may seem to contradict the formula "SSA530nm=0.801+0055*(OA:BC)" highlighted in the abstract (line 19), where a mass fraction is used.  The reviewer finds it critical that number and mass fraction are mentioned in various contexts as a proxy for SSA, while the more precise concept of a physical mixing state is not mentioned. Furthermore, the basic definitions of the quantities such as SSA and Ångström exponents should be presented before showing "first order estimates".

SSA, the absorption Ångström exponents, and MAC are now defined previously within section 2.7. We see how the use of the term 'to first order' is confusing and have replaced the phrase with 'strongly'. The reviewer makes a good point that number/mass fraction alone are not complete proxies for SSA. We do not have the data to infer mixing state but now include a paragraph discussing mixing state implications in the last section (the paragraph is included in the response to reviewer 1). We are planning to extend the current study to ORACLES-2017 and ORACLES-2018 data, for which the SP2 mixing state data are available, and will dive more deeply into how the mixing state affects the SSA.

Line 354: What plot shows the correlation between AAE and OA:BC? Is the result significant considering the small range of AAE values and typical uncertainties? This plot is not shown because the correlation is so weak. The correlation coefficient of 0.27 is not statistically significant. We now mention this. We don't mean to imply a genuine correlation and wonder if the subsequent sentence ("….does suggest that perhaps some of the secondary OA is absorbing….") could be misinterpreted. We have therefore removed that phrase.

Figure 12: The reviewer believes that there is an error in the calculation of the volume distribution. The modal diameters of the volume distribution should be larger than those of the corresponding number distribution, and the width of the volume distribution is usually equal or larger. The ratios of the total volumes also appear to be incorrect. The reviewer suspects that the total volumes for cases f44>0.18 and f44 >0.21 should be closer to case f44>0.15 than the figure shows. Accordingly, all statements referring to the figure should be verified.

We thank the reviewer for noticing this! There was indeed a basic error in the calculation of the volume distribution. This figure has been redone.

Line 405: "The heating can be interpreted as a proxy for dilution, as both physical processes increase volatility." The reviewer does not understand the content. We removed this sentence, as we describe what we take away from Fig. S13 again on the next page.

Supplement:

Line 23: The nephelometer wavelength is 550 nm. Fixed

Line 24: The SP2 derives refractory carbon rBC. See also Petzold et al. (2013) to differentiate between BC, eBC and rBC. Fixed. We cite the Petzold paper within the main text as well.

Line 24: What is the uncertainty in total aerosol mass when derived from AMS and SP2. Close to 40% for a one-minute average, as laid out in the main text. We now also include this number in this section.

Lines 22 to 31: The reviewer can not follow the method of calculating wall losses, especially what is the role of the mass scattering coefficient? We thank the reviewer for this comment, as it led us to reexamine the language. We have rewritten it as follows: The ~8 m distance from the SDI increases the potential for aerosol to deposit upon the tubing wall prior to reaching the AMS. Submicron aerosol is assumed to scatter 5 M m$^{-1}$ at a wavelength of 500 nm per $\mu$g m$^{-3}$ of aerosol mass \citep{Reid98,Haywood03}, termed by them as a mass scattering

efficiency. Their result is used to assess if submicron aerosol was lost to wall deposition before reaching the AMS during the ORACLES campaign. The aerosol scattering was measured directly behind the inlet (TSI 3563, 550 nm wavelength) and divided by the total aerosol mass ascertained by the AMS. If the mass becomes depleted by wall losses, the mass scattering efficiency becomes increased. This ratio was evaluated at three different locations/altitudes, resulting in an average scattering by submicron aerosol was 5.92  M m$^{-1}$ per $\mu$g m$^{-3}$. This is close to the expected value of 5, given that the uncertainty in the total aerosol mass concentration is almost 40%. This result constrains the wall losses to within 20% for the entire campaign, though with a wide error margin.

Line 87: Should be Figure S6? Yes. We have rechecked all the other figure numbers as well.

Line 90ff: The Anderson and Ogren (1998) correction is on correcting the nephelometer and not to derive absorption coefficients from PSAP. Oof, true. We should have caught that earlier. Thank you. Now corrected. What method was used for the PSAPs? It is not clear how the Virkkula (2010) correction is used. The Virkkula corrections are applied to the PSAP absorption coefficients prior to calculating the SSA. We have clarified this in the text.

The reviewer suggests discussing the nephelometer first, since this instrument is used for PSAP correction and subsequently for the derivation of SSA. Done.

Figure S5: Were UHSAS diameters corrected for refractive index? Yes

Line 55: It is unfortunate to call the combination of instrument (LDMA & CPC) for measuring the particle number size distribution LDMA. The TSI 3934 should be named as SMPS (scanning mobility particle sizer). The LDMA is not a TSI 3934. It was taken from a TSI 3071A electrostatic classifier and heavily modified. The flow control, neutralizer and high voltage systems have all been replaced. Although the TSI 3081 DMA is of a more recent vintage, the HiGEAR LDMA column appears to be the same (from what we can tell). The initial system was modified to scan the voltage, making it similar to an SMPS, but the original nomenclature has remained. We now include this information in the supplement.

Figures S1 and S2: Many acronyms (e.g. UCN, ACN, RCN, RRwet, RRdry, … ) are not explained. This has been corrected.

S6,  figure caption: "OA > 20 $\mu$g/m3" fixed

S4, figure caption: What is the color scale showing? CO in units of ppmv, with zero background CO. This color scale has been redone to make it more readable, including placing the CO concentrations in units of ppbv.

References

Petzold, A., Ogren, J. A., Fiebig, M., Laj, P., Li, S.-M., Baltensperger, U., Holzer-Popp, T., Kinne, S., Pappalardo, G., Sugimoto, N., Wehrli, C., Wiedensohler, A., and Zhang, X.-Y.: Recommendations for reporting "black carbon" measurements, Atmos. Chem. Phys., 13, 8365–8379, https://doi.org/10.5194/acp-13-8365-2013, 2013.

New references invoked within the response:

Adebiyi, A. A., Zuidema, P., Chang, I., Burton, S. P., and Cairns, B.: Mid-level clouds are frequent above the southeast Atlantic stratocumulus clouds, Atmos. Chem. Phys., pp. 11 025–11 043, https://doi.org/10.5194/acp-20-11025-2020, 2020.

Chakrabarty, R. K. and Heinson, W. R.: Scaling Laws for Light Absorption Enhancement Due to Nonrefractory Coating of Atmospheric Black Carbon Aerosol, Phys. Rev. Lett., 121, https://doi.org/10.1103/PhysRevLett.121.218701, 2018.

Dang, C., Segal-Rozenhaimer, M., Che, H., Zhang, L., Formenti, P., Taylor, J., Dobracki, A., Purdue, S., Wong, P.-S., Nenes, A., Sedlacek, A., Coe, H., Redemann, J., Zuidema, P., and Haywood, J.: Biomass burning and marine aerosol processing over the southeast Atlantic Ocean: A TEM single particle analysis, Atmos. Chem. Phys., https://doi.org/10.5194/acp-2021-724, 2022.

Forrister, H., Liu, J., Scheuer, E., Dibb, J., and et al., L. Z.: Evolution of brown carbon in wildfire plumes, Geophys. Res. Lett., 42, 4623–4630, https://doi.org/10.1002/2015GL063897, 2015.

Lee, J. E., Gorkowski, K., Meyer, A., Benedict, K., Aiken, A. C., and Dubey, M. K.:Wildfire smoke demonstrates significant and predictable black carbon light absorption enhancements, Geophys. Res. Lett., 49, https://doi.org/10.1029/2022GL099334, e2022GL099334, 2022.

Liu, D., Whitehead, J., Alfarra, M. R., Reyes-Villegas, E., Spracklen, D. V., Reddington, C. L., Kong, S., Williams, P. I., , Ting, Y.-C.,Haslett, S., Taylor, J., Flynn, M. J., Morgan,W. T., McFiggans, G., Coe, H., and Allan, J. D.: Black-carbon absorption enhancement in the atmosphere determined by particle mixing state, Nat. Geosci., 10, 184–188, https://doi.org/10.1038/ngeo2901, 2017.

---

## Author Comment (AC2)

A second review led to requests for additional clarifications. The reviewer comments are shown in black, with the author responses shown in blue and any edited manuscript language shown in italicized blue font.

The reviewer recommends the manuscript for publication after considering the following points. The uploaded manuscript reflects the updated language contained herein.

Line 115: Are trajectories shown?

Backtrajectories are shown in Fig. 10. We have changed the sentence on line 115 to make clear that we show them ("*Backtrajectories based on the HYSPLIT model \citep{Stein15}, also driven by the NCEP GFS meteorology, further illuminate the pathway the BBA traveled prior to its sampling on 24 September 2016 (shown later in Fig. 10).* ").

Lines 116-118: Why is the time for vertical transport not accounted for? Can you explain.

The NCEP back trajectories rely on the GFS large-scale vertical velocity. Their altitude change over the three days prior indicates the smoke has descended to where it was detected by the plane, shown below for the trajectories in Fig. 10. We do not account for the time that it takes the smoke to travel from the surface to altitudes of 3.5-6km. Fig. S3, based on the time estimate from model-released tracers, suggests it might take about 3 days for the smoke to reach the higher altitudes above the surface fire emission sources, but this is likely a model-dependent value. WRF-AAM places all the smoke in the surface layer and then lets the model advect the smoke upwards.

[Figure]

We have expanded the language around lines 116-118 to now read as: "*These backtrajectories reach the location of the fire emission source after approximately three days, at higher altitudes than where the aerosol was sampled. The aerosol age estimate is younger than the WRF-AAM aerosol forecasts would suggest. This is because the time needed for the aerosol to travel from*

*the surface to the higher altitude is not accounted for in the HYSPLIT age estimate. The vertical advection timescale is highly model-dependent on boundary layer parameterizations and we merely make note of the difference in the two age estimates here.*"
Is this clear enough?

Line 138: A detection of 99% of the BC mass is relatively high. Does this value include results from fitting the BC size distribution taking into account the calibrated SP2 detection efficiency? Is this value the same for all considered periods.

This and a latter comment by the same reviewer motivated us to dig out the full SP2 size distribution data (the campaign archived dataset only contained the total SP2 number and mass concentrations). We thank the reviewer for pushing us to do this. We now show the full number and mass distributions for the level legs detailed in Table S2 as the new figure 3, shown to the left below. Log-normal fits (examples shown in the new Fig. S4 and included here below right) indicate the upper limit of the SP2 size range - 0.5 micron - captures the upper end of the size range. At the lower end, the drop off in detection sensitivity for diameters < 0.08 micron is visible in the number size distributions for the 8/31/2016, 9/24/2016 and 9/25/2016 sample. Also evident though is that the contribution to the total mass at these small sizes is small - close if not quite at the 99% number reported within Taylor et al., 2020 - and can basically be neglected. Because of the instrument artifacts visible in the mass size distributions below, the difference in the measured mass and mass calculated from the log-normal fit was not dominated by the drop-off in detection efficiency for the smaller sizes and we do not report those numbers - but they are close to each other.

[Figure]

We have rewritten the language for this section as: "*Size distributions by number and mass (the latter assuming a BC density of 1.8 g cm$^{-3}$ \citep{Bond06}) for the level legs detailed in Table S2 indicate particle diameters that are well below the upper SP2 detection limit (Fig.~\ref{rBC_dist}; the kink at 2.5 $\mu$m for the 2016 samples is attributed to an instrument artifact). Lognormal fits help visualize a drop-off in detection efficiency for diameters < 0.08 micron for the samples weighted towards the smaller sizes (Fig. S4, for 9/24/2016 and 9/25/2016). The SP2 size range nevertheless captures almost all of the black carbon mass, close to the 99\% value reported for CLARIFY \citep{Taylor20}. rBC total number concentrations vary between 530 to 1370 cm$^{-3}$ (Fig.~\ref{rBC_dist}a), and undercounting of the mass and number through coincidence is estimated to be less than 3\% based on \cite{Taylor20}.* "

Line 162: The reviewer thinks that the lower plot in Fig. S4 is important enough to be shown in the main text, because the selection of the analyzed periods (OA>20$\mu$g/m3) is based on OC:BC versus OA.

Fig. S4b is now Fig. 4.

Line 165: The authors refer to figures S11 and S13. It is difficult to estimate how the threshold of 10 $\mu$g/m3 affects the results without seeing the same figures for 20 $\mu$g/m3 next to it.

These figures can be compared next to each other, either on a screen or as a printout. The trends (or lack of) are the same in OA:BC as a function of f44, the chemical aging marker, for both thresholds. We did find a slight difference in BC:CO between the two thresholds for the data from 8/31/2016 and the stricter threshold led us to reject the 8/31/2016 from consideration (we had originally hoped to include it but there also wasn't enough data available with the stricter threshold). We primarily include the figures in response to an earlier reviewer. We did change the language here to say: "*The BC:$\Delta$CO and OA:BC ratios are shown for individual flights as a function of {\it f44} for both thresholds in Figs. S10-S13 in the interest of full documentation.* "

Line 189: SEA is not defined.

Thanks for catching, fixed.

Line 205: The authors arguing with the BC size distribution to explain differences between LDMA and UHSAS. Is there are reference to BC size distributions?

We show the BC size distributions now in Fig. 3. These confirm that the larger particles detected by the LDMA/UHSAS, in the 0.15-0.2 micron range, are more likely to be the ones containing the black carbon. We have added a sentence here: "*Two particle populations emerge: one with diameters between 0.15-0.2 $\mu$m that is more likely to contain black carbon, (see Fig.~\ref{rBC_dist}), and another with median diameters $<$ 0.1 $\mu$m, speculated to consist mostly of OA.*".

Line 213: Uncertainty of 30% in number or volume?

Number. Now mentioned in the text.

Line 215: Should it be 'The black carbon core mass-median diameter is also estimated using the SP2-provided mass size distribution …' ?

Yes. Now that we have the full SP2 size distribution we can say this. Since we emphasize the SP2 size distribution earlier, we now just state "*The black carbon core mass-median diameter is used to infer fire conditions at the source*".

Line 229: Specify where it can be found in the supplement.

We now mention '*Section 7 of the Supplement*' instead of 'Supplement' on line 229.

Line 311: The reviewer believes that Fig. 7a does not support the conclusion that MAC is correlated to FrBC.

We have rewritten the sentence as '*The bulk mass absorption coefficients (MAC$_{BC,660}$) decrease with the estimate of the fraction of particles containing black carbon (FrBC) for the ORACLES-2016 flights (Fig. 7a), with the SSA values decreasing more systematically with FrBC for all four flights (Fig. 7b).*'

Line 337: How was TC measured?

TC is calculated as TC=BC+organic carbon (OC) and OC is estimated from OA:OC=1.26*O:C+1.18 (Aiken et al., 2008), same as is done within Brown et al., 2021. We included this information within the caption of Fig. 8. We have moved this information to the main text based on this comment.

Line 410: Is there any evidence in the literature that a significant mass loss by this mechanism is possible?

The reviewer is pointing to mass loss by oxidation through fragmentation. Jimenez et al., 2009 and O'Brien and Kroll, 2019, both cited, were influential in our thinking. We are only speculating -- and certainly if there is a better idea we would love to hear it. We do not detect much brown carbon and are looking for explanations that do not involve brown carbon. Fig. 12 makes clear that mass loss is genuinely occurring

Figure S1: Color scales not correct or overlap with other scales

I think the reviewer is referring to the top left hand figure, for which the total aerosol optical thickness color scale differs from that of the other aerosol forecasts. This was at the very beginning of the ORACLES-2016 campaign and the forecasts were still under development. This image is all we have.

Supplement line 115: The Anderson and Ogren (1998) correction is used only to correct the nephelometer.

Correct. But this is already stated in the supplement.

Supplement line 116: Heating to 50 °C can lead to a loss of volatile material. This could reduce the light enhancement factor by making the coating thinner. In addition, the mass loss could affect the correction of the light scattering artifact when using the Virkkula (2010) correction. Does this significantly affect the derived light absorption coefficient and the single scattering albedo?

As shown in Fig. S6, the size TDMA size distributions differ little between unheated samples and samples heated to 150C. The lack of volatility is consistent with the OA regimes shown in Fig. 6 (the ORACLES aerosol are primarily classified as 'aged' and 'semi-volatile' regimes). Barrett et al., 2022 (cited) compared the ORACLES PSAP absorption values to the more accurate absorption measurements by the CLARIFY EXSCALABAR and DOE CAPS-SSA instrument, and these corresponded well. Based on these various independent assessments we claim that any loss of additional volatile material by heating to 50C over a  short time period will fall within the measurement  uncertainties.